# Trypsin is a coordinate regulator of N and P nutrients in marine phytoplankton

Yanchun You[1], Xueqiong Sun[1], Minglei Ma[1], Jiamin He[1], Ling Li[1], Felipe Wendt Porto[2] & Senjie Lin [1,2✉]

Trypsin is best known as a digestive enzyme in animals, but remains unexplored in phytoplankton, the major primary producers in the ocean. Here we report the prevalence of trypsin genes in global ocean phytoplankton and significant influences of environmental nitrogen (N) and phosphorus (P) on their expression. Using *CRISPR/Cas9* mediated-knockout and overexpression analyses, we further reveal that a trypsin in *Phaeodactylum tricornutum* (*PtTryp2*) functions to repress N acquisition, but its expression decreases under N-deficiency to promote N acquisition. On the contrary, *PtTryp2* promotes phosphate uptake per se, and its expression increases under P-deficiency to further reinforce P acquisition. Furthermore, *PtTryp2* knockout led to amplitude magnification of the nitrate and phosphate uptake 'seesaw', whereas *PtTryp2* overexpression dampened it, linking *PtTryp2* to stabilizing N:P stoichiometry. Our data demonstrate that *PtTryp2* is a coordinate regulator of N:P stoichiometric homeostasis. The study opens a window for deciphering how phytoplankton adapt to nutrient-variable marine environments.

[1] State Key Laboratory of Marine Environmental Science, College of Ocean and Earth Sciences, Xiamen University, Xiamen, Fujian 361102, China.
[2] Department of Marine Sciences, University of Connecticut, Groton, CT 06340, USA.  ✉email: senjie.lin@uconn.edu

Trypsin (EC 3.4.21.4) is a proteolytic enzyme that cleaves polypeptides specifically at the carboxyl end of the lysine and arginine residues. As a large family of enzymes, trypsin is structurally and functionally conserved from bacteria to mammals[1] but is believed to be absent in plants and protists[2]. In animals, trypsin is best known as a digestive enzyme, digesting protein food or activating other proteases for digestion[3,4]. In a recent study, the trypsin gene was found to be expressed at an extremely high level (1% of the total diatom transcriptome) in diatoms that dominated the phytoplankton community during a regime shift, in which a dinoflagellate bloom was emerging and phosphate was sharply declining[5]. This finding prompted us to ask whether trypsin occurs and functions in nutritional regulation in phytoplankton, the major contributor of marine biodiversity and global $CO_2$ fixation and $O_2$ production.

In this work, by mining existing genomic and transcriptomic data, we find wide occurrence and expression of trypsin genes in phytoplankton in the global ocean, and strong responses of Bacillariophyta and Chlorophyta trypsins to environmental stimuli, particularly the variation of N and P nutrients. By physiological, molecular, and functional genetic analyses, we further unveil the function of a diatom trypsin gene as a coordinate regulator of N and P, the two major nutrients that control marine phytoplankton productivity.

## Results and discussion
**Widespread occurrence and environmental stimuli responsiveness of trypsin in marine phytoplankton.** To assess whether trypsin occurs broadly in marine phytoplankton and what ecological functions phytoplankton trypsin genes may play, we investigated the occurrence of trypsin genes and environmental stimuli regulating their expression based on PhyloDB, *Tara* Oceans unigenes and metatranscriptomes datasets. From *Tara* Oceans unigenes and metatranscriptomes, trypsin homologs were found at all the sampling stations worldwide and in all major phytoplankton phyla (Fig. 1a and Supplementary Fig. 1). The broad phylogenetic representation is corroborated by the prevalence of trypsin in the individual species' transcriptomes in the PhyloDB database (Fig. 1b), most notably in Bacillariophyta, Dinophyta, Chlorophyta, Cryptophyta and Haptophyta, the major eukaryotic groups of phytoplankton in the ocean. These indicate that trypsin is widely distributed in phytoplankton both taxonomically and geographically, a finding that advances our knowledge on the distribution of this ancient enzyme. Moreover, phylogenetic and structure alignment analysis showed that phytoplankton trypsins are more closely related with bacterial trypsins than metazoan and fungal counterparts, but contain the conserved important residues and structure typical of animal trypsins (Supplementary Figs. 2–4). These observations suggest some complex evolutionary trajectory that might result in functional innovation of phytoplankton trypsin.

We found a large amount of trypsin gene duplication, 5 copies to 65 copies in each algal genome we examined[6]. The evolution of the gene family, in gene sequence and organization relative to other functional domain, need to be treated in a separate paper[6], but the rampant gene duplication suggests that trypsin may have important roles in phytoplankton. Moreover, our correlation analysis for trypsin gene expression with environmental parameters in the *Tara* Oceans metatranscriptomic data showed that the phytoplankton trypsin transcript abundance was correlated with environmental conditions in some taxa, size fractions, and water depths, evidence that trypsin may be important in phytoplankton to adapt to dynamical environmental conditions[6]. To further explore specific environmental drivers modulating the expression of trypsin, we analyzed distance-corrected dissimilarities of phytoplankton trypsin transcript abundance with environmental nutrient factors using the partial Mantel test. Analyses were restricted to the 5–20 and 20–180 μm size fractions from surface layer as their trypsin appeared to be more responsive to environmental stimuli. As shown in Fig. 1c, trypsin expression in Bacillariophyta, Dinophyta, Chlorophyta, Cryptophyta and Haptophyta was differentially correlated with nutrient availability, most notably in Bacillariophyta and Chlorophyta. Moreover, nitrate and nitrite (NO3, NO3_5m*, and NO3_NO2) and phosphate (PO4) were the strongest correlates of both Bacillariophyta and Chlorophyta trypsin transcript abundances (Fig. 1c). Hence, we posit that trypsin have important functions in the response of phytoplankton to N and P nutrient conditions.

**Involvement of trypsin in nitrogen and phosphorus nutrient responses.** To gain mechanistic insights into the function of trypsin in phytoplankton, we conducted experiments on the model diatom *Phaeodactylum tricornutum*. We identified ten trypsin genes from its genome (Supplementary Table 1), and based on qRT-PCR, we observed their growth stage- and condition-specific expression variations (Fig. 2a and Supplementary Fig. 5). Interestingly, one of these genes (*PtTryp2*) exhibited opposite directions of expression dynamic under N- and P-depleted conditions: downregulated under N-depleted but upregulated under P-depleted condition (Fig. 2a). Furthermore, *PtTryp2* transcript increased with increasing cellular N content but decreased with increasing cellular P content (Fig. 2b, c). These results suggest that *PtTryp2* is involved in an opposite-direction regulation of responses to nitrogen and phosphorus nutrient status.

To interrogate the function of *PtTryp2* in N and P nutrient responses, we analyzed the physiology of homologous over-expression and *CRISPR/Cas9* knockout lines we generated. A *PtTryp2*-overexpression cell line with C-terminal *eGFP* fusion (named *PtTryp2*-OE) was generated, and the expression of OE cell line was confirmed at a protein level through Western blot (Fig. 3a). Because the function of a protein corresponds with its subcellular location, we first examined where *PtTryp2* is located inside *P. tricornutum* cells. By computational simulation, we find *PtTryp2* is potentially localized in the chloroplast via the secretory pathway (Supplementary Table 2), in accordance with the fact that chloroplasts contain a rather high number of proteases and are the main location of nutrients assimilation and remobilization[7]. To obtain experimental verification of the chloroplast localization, we carried out subcellular localization analysis in the OE and OEC cell lines using confocal fluorescence microscopy. Interestingly, results show *PtTryp2-eGFP* are localized in both the chloroplast and cytoplasmic endoplasmic reticulum (ER), to the exclusion of the nucleus and Golgi apparatus, whereas the fluorescence from the *eGFP* blank vector control is outspread in the cell instead of being co-localized with chloroplast and ER (Fig. 3b and Supplementary Figs. 6–8). Further analyses show that *PtTryp2* lacks the C-terminal -(K/H) DEL sequences, a typical ER-retention signal that prevents ER-resident proteins from being transported to downstream locations of the secretory system[8,9]. Hence, *PtTtryp2* is evidently transported via the ER to the chloroplast, as in the case of the previously documented light-harvesting chlorophyll a/b-binding protein in *Euglena*[10].

*PtTryp2* contains one trypsin domain and two internal repeats 1 (RPT) (Fig. 4a), offering one single target for trypsin mutagenesis. Using an optimized efficient *CRISPR/Cas9* gene editing system[11], we obtained three *PtTryp2* mutants with different mutation characteristics in the trypsin domain (named KO1, KO2, and KO3, respectively; Fig. 4b). As shown in Fig. 4c,

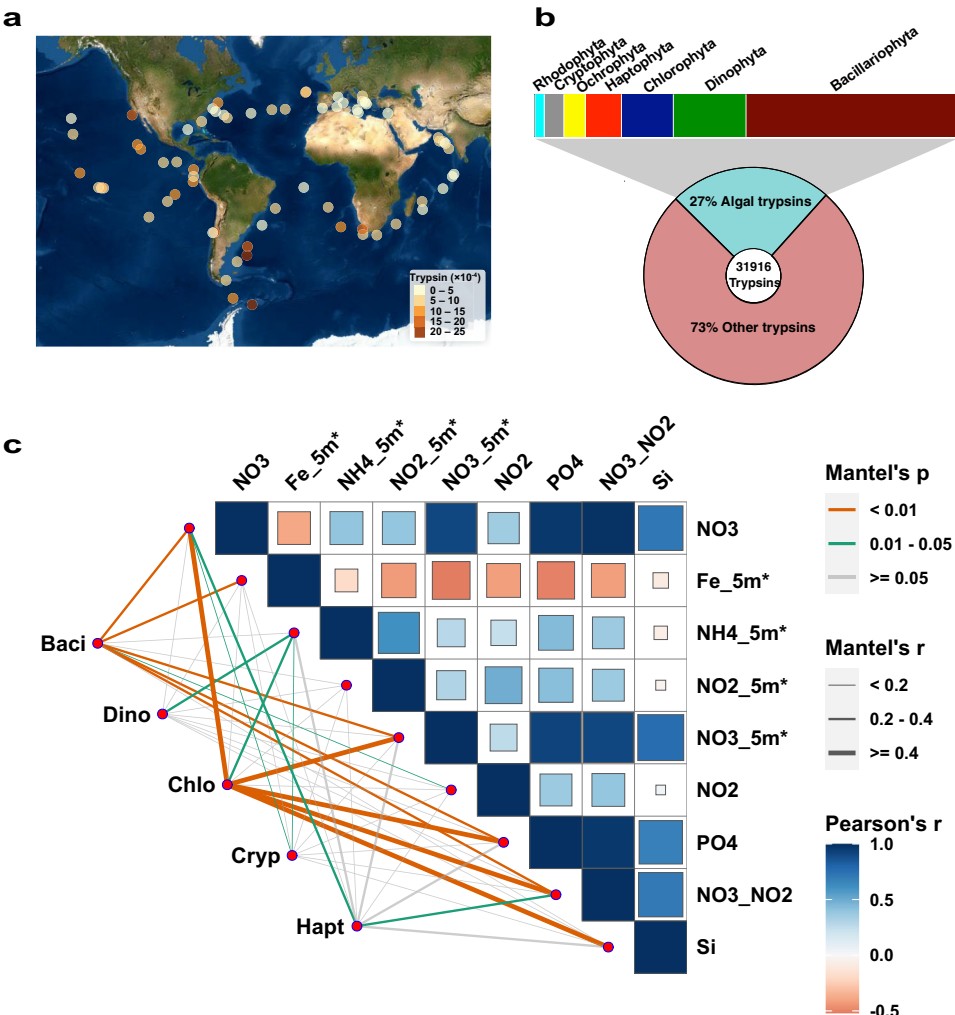

**Fig. 1 Widespread occurrence and environmental nutrient responsiveness of trypsin in global marine phytoplankton. a** Wide geographic distribution of trypsin in phytoplankton found in *Tara* Oceans. Color scale depicts trypsin mRNA abundance. **b** Wide taxonomic distribution of trypsin in algae found in PhyloDB. **c** Environmental nutrient drivers of phytoplankton trypsin abundance. Pairwise comparisons of environmental nutrient concentrations are shown with a color gradient denoting Pearson's correlation coefficient. The trypsin abundance and taxonomic distribution based on the 5–180 μm size fraction from SRF layer from *Tara* Ocean datasets. Taxonomic trypsin abundance was related to each nutrient factor by partial (geographic distance-corrected) Mantel tests. Edge width corresponds to the Mantel's *r* statistic for the corresponding distance correlations, and edge color denotes the statistical significance based on 9999 permutations. Baci Bacillariophyta, Dino Dinophyta, Chlo Chlorophyta, Cryp Cryptophyta, Hapt Haptophyta. Source data are provided as a Source Data file.

compared with the knockout control cell line (KOC), all three *PtTryp2*-KO lines exhibited a significantly diminished *PtTryp2* expression under both nutrient depletion and repletion; conversely, the OE cell line displayed markedly elevated *PtTryp2* expression in comparison to the overexpression control cell line (OEC). Moreover, the *PtTryp2* expression level in KOC cell lines strongly responded to the ambient N and P level, but consistently showed a constant and low expression pattern in KO lines (Fig. 4d). These results verified that KO cell lines with the loss of *PtTryp2* function, and OE with enhanced function of *PtTryp2*, can be used for subsequent functional analyses of *PtTryp2*.

Moreover, we observed the growth physiology of different *PtTryp2* mutants across different nutrient conditions. As shown in Fig. 4e and Supplementary Fig. 9, both of the knockout and overexpression of *PtTryp2* resulted in decreases in the exponential growth rates (days 1–4) and maximum cell density across different N and P culture conditions. Taken together, these results demonstrate that both elevation and reduction of *PtTryp2* expression result in cell growth repression, evidence that *PtTryp2*

has a crucial role in modulating cell growth in response to different N and P conditions.

***PtTryp2* represses nitrogen assimilation and metabolism.** Transcriptomic data show that *PtTryp2* knockout led to the upregulation of most of the nitrogen assimilation and metabolism genes under both N-depleted and replete conditions (Fig. 5a). The transcriptomic data are confirmed to be reproducible based on the correlation analysis of housekeeping genes (Supplementary Fig. 10 and Supplementary Table 3). Notably, the expression fold change of most N assimilation and metabolism genes under N-depleted, P-replete (LNHP) versus nutrient repete (HNHP) conditions were moderated in the *PtTryp2* knockout mutant compared to that in its control (KOC), with the exception of GOGAT, which exhibited larger response to the nutrient changes in KOC (Fig. 5a). All these indicate that the inactivation of *PtTryp2* enhanced N assimilation and metabolism to mitigate cell stress and reduce overall transcriptomic swing from N-depletion. Under replete conditions (HNHP), substantial transcriptional

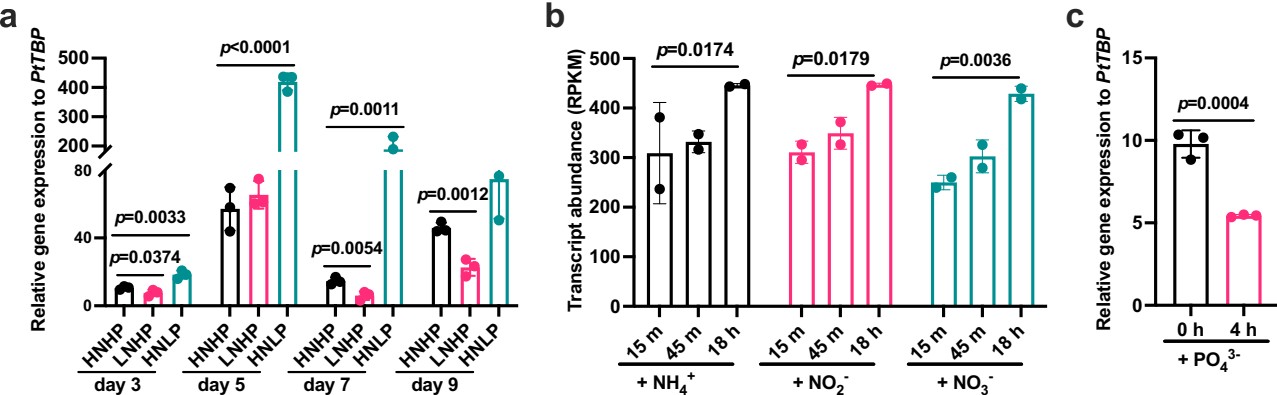

**Fig. 2 Involvement of *PtTryp2* in nitrogen and phosphorus nutrient responses. a** *PtTryp2* expression in *P. tricornutum* under different growth stages and conditions based on qRT-PCR. Nutrient-replete, HNHP; N-depletion, LNHP; P-depletion, HNLP. Data are presented as mean values ± SD ($n = 3$ biologically independent samples). The comparisons between the averages of the two groups were evaluated using the one-tailed Student's *t* test. The *p* values with significance ($p \leq 0.05$) are shown. **b** Time-course expression patterns of *PtTryp2* when *P. tricornutum* was grown with different forms of nitrogen nutrients. Data are presented as mean values ± SD ($n = 2$ biologically independent samples). The comparisons between the averages of the two groups were evaluated using the one-tailed Student's *t* test. The *p* values with significance ($p \leq 0.05$) are shown. **c** *PtTryp2* expression pattern after phosphorus supplement. Data are presented as mean values ± SD ($n = 3$ biologically independent samples). The comparisons between the averages of the two groups were evaluated using the one-tailed Student's *t* test. The *p* values with significance ($p \leq 0.05$) are shown. Source data are provided as a Source Data file.

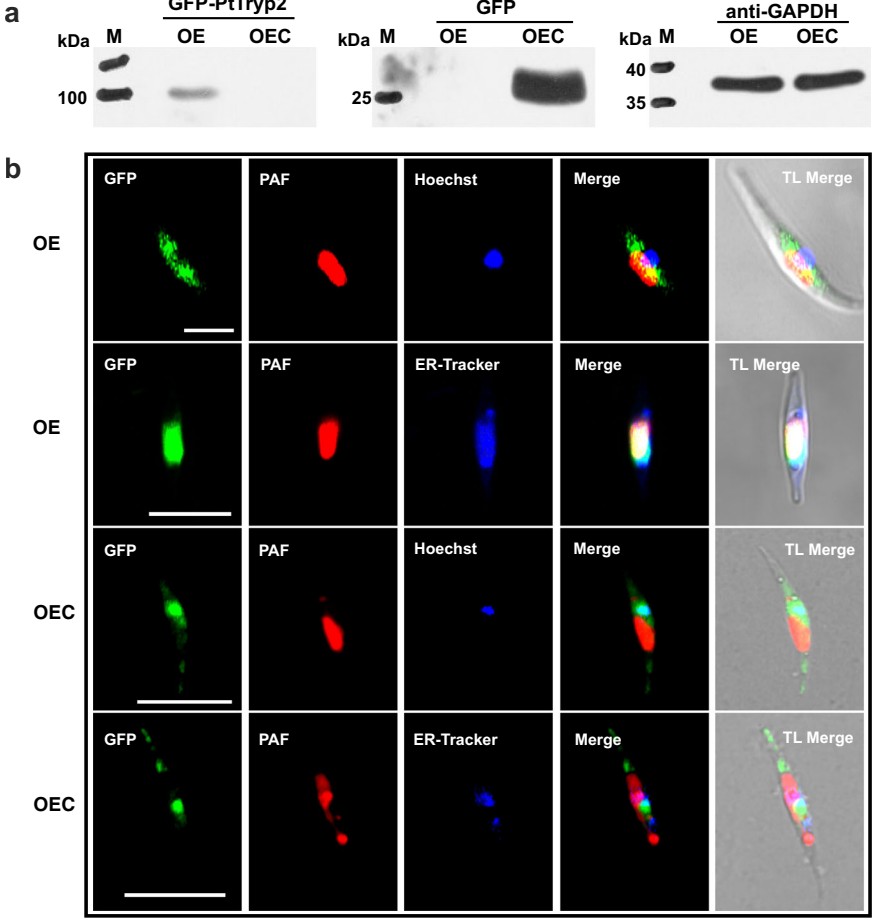

**Fig. 3 Subcellular localization of PtTryp2. a** Detection of the expression of GFP-*PtTryp2* by Western blot using anti-GFP primary antibody. Left panel, GFP-*PtTryp2* fusion protein. Middle panel, GFP protein. GAPDH (on the right) was detected using anti-GAPDH as the control to indicate equal protein quantities loaded to each lane. The GFP-*PtTryp2* was confirmed expressed successfully at protein level in OE cell line. All experiments were repeated independently three times, and similar results were obtained. **b** Confocal micrographs showing subcellular localization of GFP-*PtTryp2* in chloroplast (PAF, showing red autofluorescence) and endoplasmic reticulum (ER, showing blue fluorescent stain by ER-Tracker) but not in nucleus (Hoechst 33342, showing blue fluorescent stain). TL merge, merger of the fluorescence images with transmission light image. Scale bar, 10 μm, applies to all images. All experiments were repeated independently three times, and similar results were obtained. Source data are provided as a Source Data file.

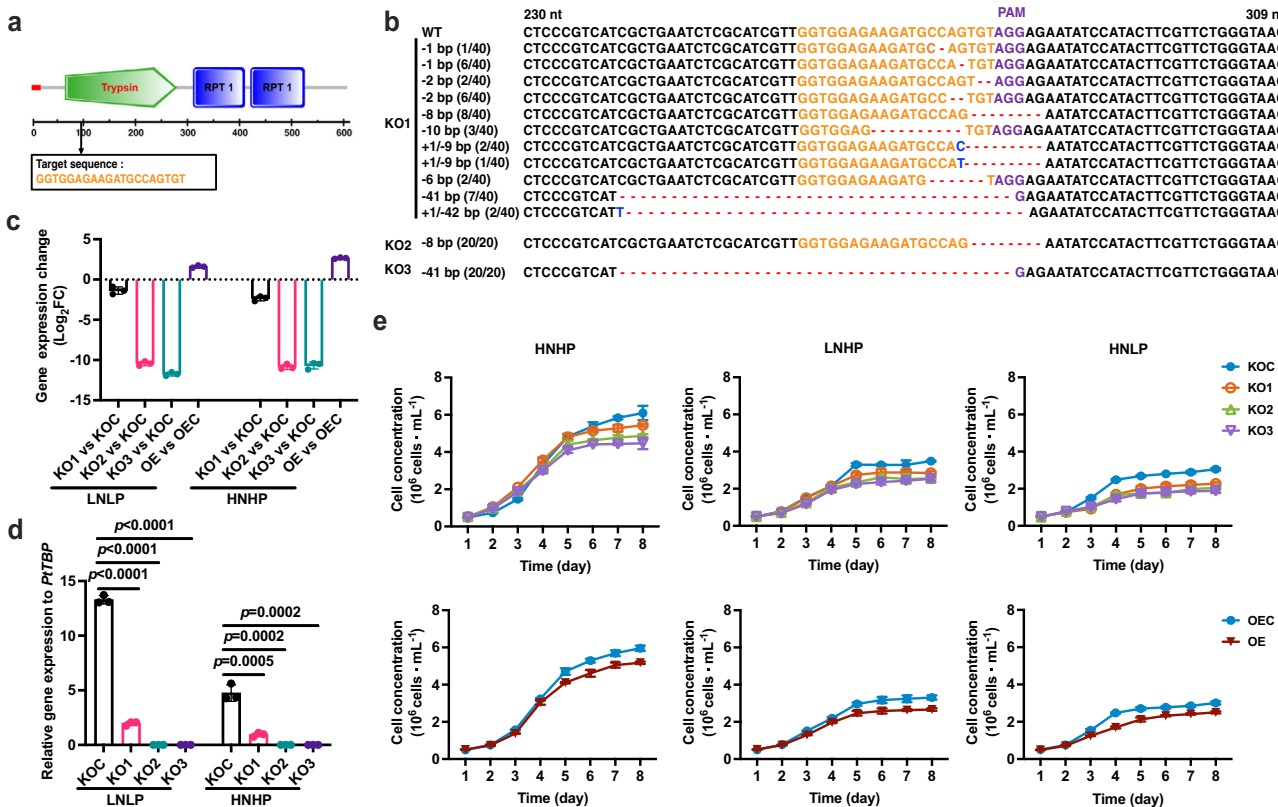

**Fig. 4 Mutation generations of *PtTryp2* and characters of mutants. a** Schematic presentation of *PtTryp2* protein. The target site (vertical arrow) for *CRISPR/Cas9*-based knockout is located within the conserved functional domain (green pentagon), with PAM motif shown in orange font. Red rectangle on the left depicts signal peptide; RPT: internal repeat 1; **b** Alignment of partial *PtTryp2* sequences of the *CRISPR/Cas9*-generated mutants showing frameshift indels compared to wild type. The frequency by which the sequence was detected within the same colony is indicated in parenthesis. Font color coding: Black, WT sequence; Orange, functional domain containing target for *CRISPR/Cas9*; Purple, PAM sequence; Blue, Inserted bases; Red dashes, deleted bases. **c** *PtTryp2* expression patterns of knockout and overexpression mutants under different conditions. FC fold change. Data are presented as mean values ± SD ($n = 3$ biologically independent samples). **d** *PtTryp2* expression of knockout mutants exhibited no response to ambient N and P fluctuation. Data are presented as mean values ± SD ($n = 3$ biologically independent samples). The comparisons between the averages of the two groups were evaluated using the one-tailed Student's *t* test. The *p* values with significance ($p \leq 0.05$) are shown. **e** Growth curves of different *PtTryp2* mutants under different N and P conditions. Nutrient conditions in **c–e** are indicated by HNHP (Nutrient-replete), LNHP (N-depleted, P-replete), HNLP (N-replete, P-depleted), and LNLP (Nutrient-depleted). Data are presented as mean values ± SD ($n = 3$ biologically independent samples). Source data are provided as a Source Data file.

reprogramming and a significant increase in nitrate uptake rate and cellular N content was observed in the knockout mutants (KO1, KO2 and KO3) (Fig. 5b). The physiological changes were reversed in the overexpression cell lines: a decline in nitrate uptake rate and cellular N content was noted in *PtTryp2*-OE (Fig. 5c). All the results demonstrate that *PtTryp2* functions as a repressor of nitrogen assimilation and metabolism.

In addition, when comparing N-depleted with N-replete conditions, 646 differentially expressed genes (DEGs) were identified in the blank vector control (KOC) but only 187 in *PtTryp2*-KO1, considerably fewer in the knockout mutant (Fig. 5d). Besides, the magnitude of change was smaller in *PtTryp2*-KO1 than in KOC for the majority (73%) of the DEGs (Fig. 5e). It is thus evident *PtTryp2* in the wild type functions as an amplifier of general metabolic response to N-starvation by repressing nitrogen assimilation and metabolism. Notably, the *PtTryp2*-KO-promoted and *PtTryp2*-OE-repressed $NO_3^-$ uptake patterns observed under nutrient repletion were reversed under P-depletion, indicating that *PtTryp2*'s roles in N and P signaling are not separated, but rather the protein might mediate the cross-talk between N and P signaling.

Besides the direction of action (repression or promotion) shown above, the function of *PtTryp2* involves another layer of

regulation: the direction of its own expression changes. We find that *PtTryp2* expression decreased under N-depletion and increased after N-supplement. Under this two-level regulatory scheme, *PtTryp2* is a repressor of N uptake and assimilation genes and a promoter of N starvation-responsiveness in general metabolic pathways per se; yet its own expression decreases under N-limitation to upgrade N-uptake and assimilation under N depletion, and increases under N richness to prevent excessive N-uptake and assimilation; meanwhile, the decreased expression of *PtTryp2* actually dampens the dynamic swing in the metabolic landscape in response to N-starvation. This *PtTryp2*-based regulatory mechanism might enable cells to swiftly respond to fluctuating N availability and cellular demand in order to finetune N responses so that N acquisition is optimized.

**PtTryp2 promotes P starvation-induced genes and Pi uptake.** As shown above, *PtTryp2* expression is downregulated under N-deficiency to release *PtTryp2*'s repressing effects on N-starvation response and to promote N uptake, thereby the cells achieve N homeostasis, and an opposite expression pattern of *PtTryp2* was observed under P-deficiency, suggesting a N-P coregulation. However, the role of *PtTryp2* in P-starvation

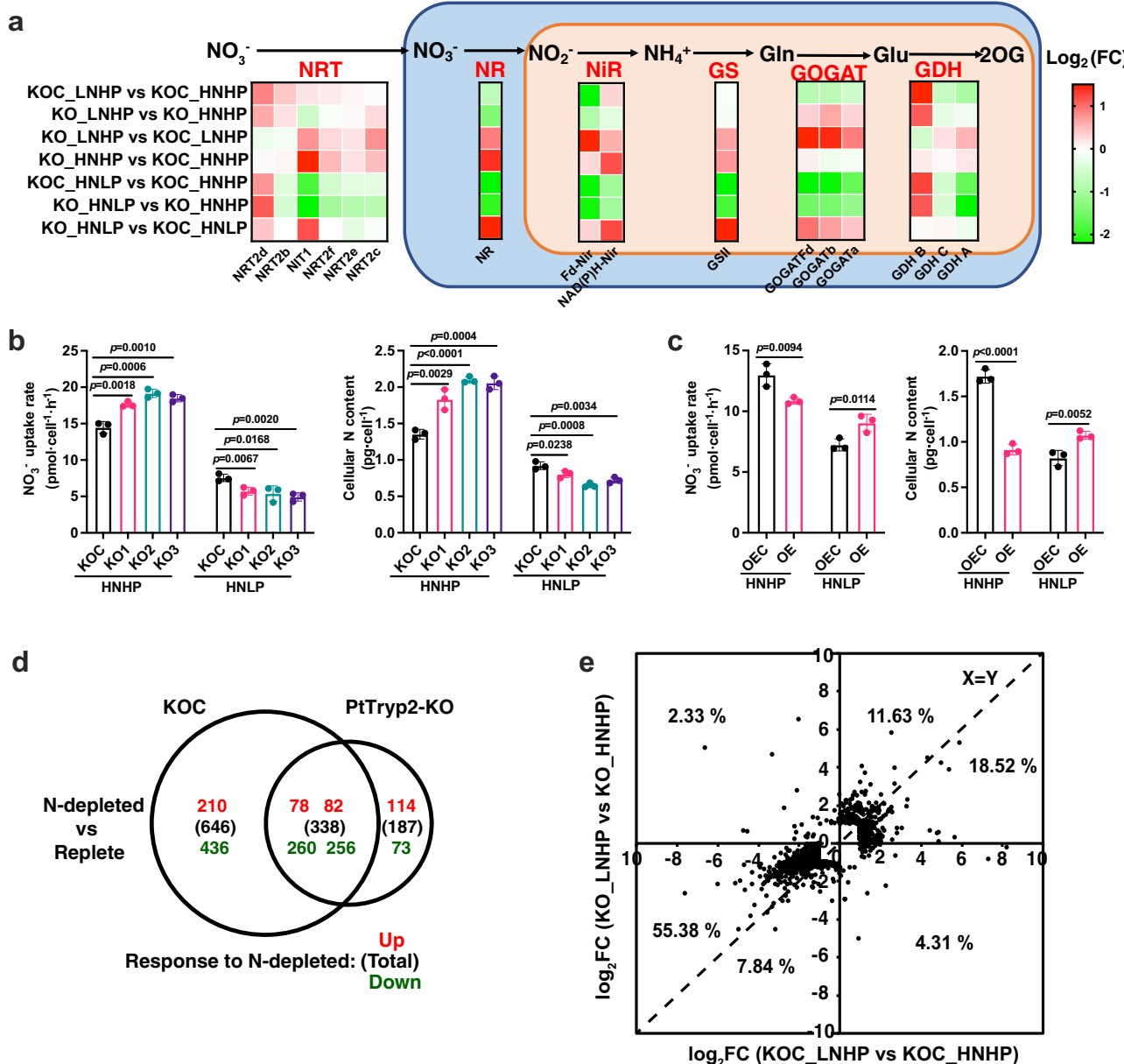

**Fig. 5 Transcriptomic and physiological evidence that *PtTryp2* directly represses nitrogen assimilation and metabolism. a** *PtTryp2* knockout resulted in upregulation of major nitrate-uptake and N-metabolism genes in *PtTryp2* knockout (KO1) and control (KOC) under N-depleted (LNHP), P-depleted (HNLP), and nutrient-replete conditions (HNHP). NRT nitrate transporter, NR nitrate reductase, NiR nitrite reductase, GS glutamine synthetase, GOGAT glutamate synthase, GDH glutamate dehydrogenase, 2OG 2-Oxoglutarate; **b** $NO_3^-$ uptake rate and cellular N content, increasing dramatically in *PtTryp2*-KO under HNHP, but decreasing remarkably under HNLP. Data are presented as mean values ± SD ($n = 3$ biologically independent samples). The comparisons between the averages of the two groups were evaluated using the one-tailed Student's $t$ test. The $p$ values with significance ($p \leq 0.05$) are shown. **c** $NO_3^-$ uptake rate and cellular N content, decreasing remarkably in *PtTryp2*-overexpressing *P. tricornutum* under HNHP, but increasing under HNLP. Data are presented as mean values ± SD ($n = 3$ biologically independent samples). The comparisons between the averages of the two groups were evaluated using the one-tailed Student's $t$ test. The $p$ values with significance ($p \leq 0.05$) are shown. **d** Venn diagram showing the number of N-depletion induced DEGs in *PtTryp2*-KO1 and KOC. In parentheses, total number of DEGs; red font, upregulated; green font, downregulated. **e** Log2 fold changes (FC) of N-depletion induced differential gene expression in *PtTryp2*-KO1 against that in KOC. Most data points (93.37%) are distributed in 1,3 quadrants, indicating the same direction of change. Source data are provided as a Source Data file.

responses and P homeostasis still needs to be unraveled. Toward that goal, we examined the effects of *PtTryp2* inactivation on the expression changes of P starvation-induced genes and the inhibitory regulator of P signaling (*SPX*), which in plants is a typical P starvation response mechanism[12]. Consistently, most of Pi transporters (PTs) and alkaline phosphatase (APs) exhibited upregulation to P starvation response in KOC, but most of *SPX* genes showed downregulation (Fig. 6a).

Interestingly, under P-depletion, *PtTryp2* knockout down-regulated the expression of most of PTs and APs, but upregulated most of the *SPX* genes (Fig. 6a), revealing *PtTryp2*'s role in WT to promote P-starvation responses. Consistent with gene transcription, *PtTryp2* knockout lowered Pi uptake rate and cellular P content under the nutrient-replete condition (Fig. 6b), whereas an increase was noted in the overexpression cell line *PtTryp2*-OE (Fig. 6c). Based on RNA-seq, remarkably more DEGs were found

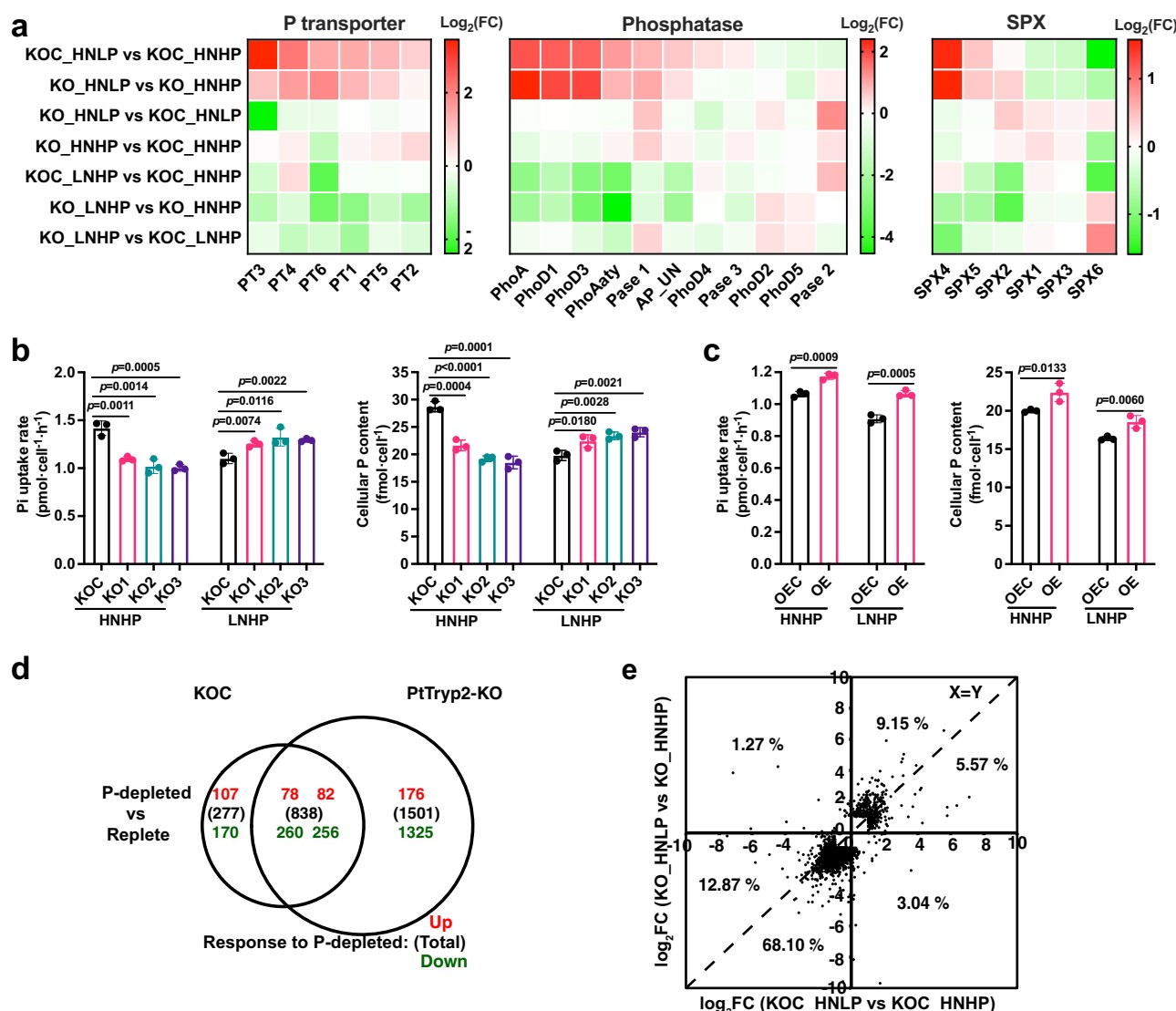

**Fig. 6 Transcriptomic and physiological evidence that *PtTryp2* positively modulates P starvation-induced genes during Pi starvation. a** *PtTryp2* knockout resulted in a reverse regulation of most P starvation-induced genes relative to that in control (KOC) under N-depleted (LNHP), P-depleted (HNLP), and nutrient-replete (HNHP) conditions. **b** *PtTryp2* knockout caused decreases in Pi uptake and cellular P content under nutrient-replete condition (HNHP) but caused increases under N-depleted condition (LNHP). Data are presented as mean values ± SD ($n = 3$ biologically independent samples). The comparisons between the averages of the two groups were evaluated using the one-tailed Student's *t* test. The *p* values with significance ($p \leq 0.05$) are shown. **c** *PtTryp2* knockout caused increases in Pi uptake rate and cellular P content under HNHP and LNHP. Data are presented as mean values ± SD ($n = 3$ biologically independent samples). The comparisons between the averages of the two groups were evaluated using the one-tailed Student's *t* test. The *p* values with significance ($p \leq 0.05$) are shown. **d** Venn diagram showing the number of P-depletion induced DEGs in *PtTryp2*-KO1 and KOC. In parentheses, total number of DEGs; red font, upregulated; green font, downregulated. **e** Log$_2$ fold changes (FC) of P-depletion induced differential gene expression in *PtTryp2*-KO1 against that in KOC. Most data points (95.69%) are distributed in 1,3 quadrants, indicating the same direction of change. Source data are provided as a Source Data file.

for the P-depleted versus nutrient-replete comparison in *PtTryp2*-KO1 (1501) than that in KOC (277) (Fig. 6d). Besides, in *PtTryp2*-KO1, the majority of these DEGs (77.25%) exhibited greater fold changes than that in KOC (Fig. 6e). These results indicate that *PtTryp2* upregulation in the wild type would dampen metabolic reprogramming in responses to P-limitation, and *PtTryp2* downregulation would prevent cells from over P accumulation after P supplement, as opposed to the response to N-depletion. All these findings are indicative that *PtTryp2* in the WT functions to upregulate the P starvation-induced genes and restrict general metabolic reconfiguration in response to P-limitation, a mechanism to maintain P homeostasis. Similar to that the *PtTryp2*-KO-promoted and *PtTryp2*-OE-repressed NO$_3^-$ uptake

patterns were reversed under P-depletion, the *PtTryp2*-KO-repressed Pi uptake pattern was reversed under N-depletion (Fig. 6b), implying that *PtTryp2* might mediate the cross-talk between N and P signaling. The *PtTryp2*-OE-promoted Pi uptake pattern was not reversed under N-depletion, however, because N-depletion downregulated the expression of *PtTryp2*, resulting in the *PtTryp2* expression pattern between OEC and OE similar to that under nutrient repletion.

**_PtTryp2_ coordinately regulate N and P uptake and mediates N-P cross-talk.** Given the *PtTryp2*-mediated cross-talk between N and P signaling in *P. tricornutum* implied in the results presented

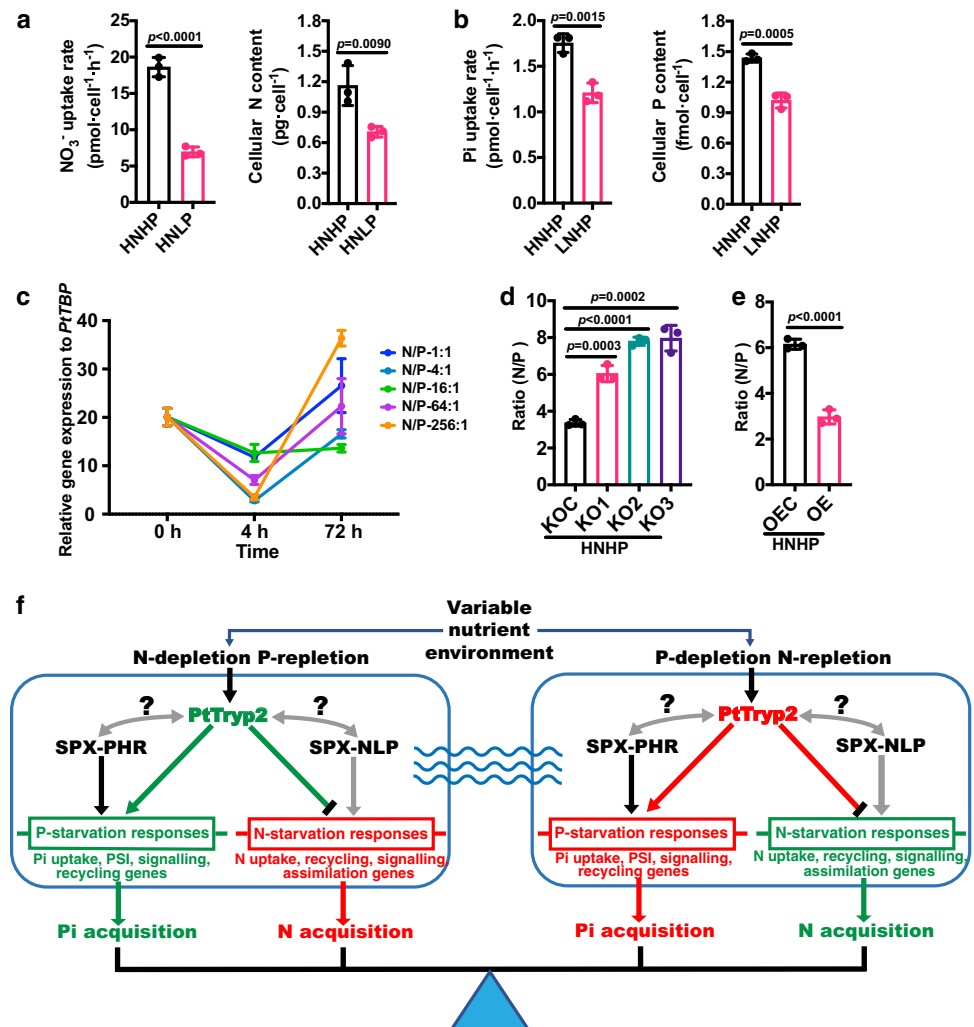

**Fig. 7 Illustration that *PtTryp2* coordinately regulates N and P acquisition under fluctuating nutritional conditions. a** $NO_3^-$ uptake and cellular N content repressed under HNLP in wild-type cells (WT). Data are presented as mean values ± SD ($n = 3$ biologically independent samples). The comparisons between the averages of the two groups were evaluated using the one-tailed Student's $t$ test. The $p$ values with significance ($p \leq 0.05$) are shown. **b** Pi uptake and cellular P content repressed under LNHP in wild-type cells (WT). Data are presented as mean values ± SD ($n = 3$ biologically independent samples). The comparisons between the averages of the two groups were evaluated using the one-tailed Student's $t$ test. The $p$ values with significance ($p \leq 0.05$) are shown. **c** Time-course expression of *PtTryp2* showed co-varied with the N/P nutrient ratio. Moreover, *PtTryp2* expression fluctuated less at the N/P ratio of 16:1 compared to other N/P ratios. The 4 h after nutrient addition represents nutrient-repletion and 72 h nutrient-depletion. Data are presented as mean values ± SD ($n = 3$ biologically independent samples). **d** The cellular N/P ratio was significantly elevated by the inactivation of *PtTryp2*. Data are presented as mean values ± SD ($n = 3$ biologically independent samples). The comparisons between the averages of the two groups were evaluated using the one-tailed Student's $t$ test. The $p$ values with significance ($p \leq 0.05$) are shown. **e** The cellular N/P ratio was significantly decreased by the overexpression of *PtTryp2*. Data are presented as mean values ± SD ($n = 3$ biologically independent samples). The comparisons between the averages of the two groups were evaluated using the one-tailed Student's $t$ test. The $p$ values with significance ($p \leq 0.05$) are shown. **f** Hypothetical model depicting the role of *PtTryp2* in balancing N and P acquisition. Under N-depletion, *PtTryp2* expression is downregulated to promote N-starvation responses and repress P-starvation responses. In contrast, under P-depletion, *PtTryp2* expression is upregulated to reinforce P-starvation responses and lessen N-starvation response. By this feedback loop, optimal N-P uptake is achieved to maintain stoichiometric homeostasis. Upregulated genes and enhancement processes are shown in red, downregulated genes and weakened processes colored green. The black arrows depict transcriptional activation. Black bar at line's end depicts inhibitory regulation. The gray arrows depict possible but unverified interaction between *PtTryp2* and the existing P regulating cascade *SPX-PHR* or an equivalent of the N regulating cascade known in plants (*SPX-NLP* where *NLP* stands for NIN-like protein, a transcription factor). Source data are provided as a Source Data file.

above, we were tempted to investigate the nature and the mechanism the cross-talk. Here, we uncover Pi and $NO_3^-$ antagonistic interactions in *P. tricornutum*, which resemble that in land plants to achieve an optimal N-P nutrient balance[13,14]. In wild-type (WT) *P. tricornutum*, we observed a significant repression of $NO_3^-$ uptake under P starvation and a significant repression of Pi uptake rate under N starvation. Consequently, cellular N content decreased under the P-depleted condition, and

cellular P content decreased under the N-depleted condition, relative to nutrient-replete conditions (Fig. 7a, b). In accordance, the transcription of N assimilation and metabolism genes was repressed by P deficiency, and that of P starvation-induced genes was repressed by N limitation (Supplementary Fig. 11). Moreover, transcriptomic results demonstrated that *PtTryp2* knockout led to the magnification of Pi and $NO_3^-$ antagonistic interaction (Supplementary Fig. 11), linking *PtTryp2* inactivation to the

disruption of the N-P homeostasis. Taken together, our data reveal that *PtTryp2*'s function operates in opposite directions for N and P responses, but in a coordinated manner, consistent with a role to coregulate N and P signaling.

To further illustrate this, we have carried out *PtTryp2* expression pattern analysis across different N/P nutrient stoichiometric ratio conditions, and found that *PtTryp2* expression co-varied with the N/P nutrient ratio (Fig. 7c). The time-course analysis showed that *PtTryp2* expression fluctuated less under different N or P conditions at the N/P ratio of 16:1 compared to other N/P ratios. The N/P nutrient ratio of 16:1 is considered balanced stoichiometry (Redfield ratio) and appears to be optimal for *P. tricornutum* growth (Supplementary Fig. 12), as previously documented[15], suggesting that at this nutrient stoichiometry there is no need for a significant change in *PtTryp2* expression to maintain N/P balance, but other N:P nutrient ratios deviating from 16:1 caused changes in *PtTryp2* expression to maintain N/P balance. Moreover, the extent of change in *PtTryp2* expression varied between cultures with different levels of N:P nutrient ratios, and between 4 and 72 h after culture inoculation from N- and P-depletion-acclimated parent culture into the experimental nutrient conditions. At 72 h *PtTryp2* expression level increased with the degree of P stress (the higher the N:P ratio, the more P stressed the cultures were), except for the N:P = 1:1 condition, an extreme N-limited condition that seemed to cause *PtTryp2* expression not to respond according to the general trend. Overall, all these data indicate that *PtTryp2* responds strongly to the variability of the N:P ratio. Correspondingly, the cellular N/P ratio under nutrient-repletion also seems to be influenced by *PtTryp2* expression level: the cellular N/P ratio was significantly elevated by *PtTryp2* knockout, but conversely, was significantly decreased by the overexpression of *PtTryp2* (Fig. 7d). Evidently, *PtTryp2* serves to coordinate N and P uptake and metabolism to dampen the amplitude of N:P ratio changes that occur when the *P. tricornutum* cells experience fluctuations in nutrient conditions[16,17]. That is, *PtTryp2* in *P. tricornutum* acts like an amplitude reducer of the N-P seesaw to achieve the N and P stoichiometric homeostasis (Fig. 7f).

As critical nutrients for phytoplankton and plants, the balance and homeostasis of N and P are crucial to the growth of the organisms. For plants, nutrient supply in the soil is highly variable; therefore, to achieve optimal and coordinated utilization of N and P, integration of N and P signaling into an integrated network is required[18]. Recent studies have revealed the critical components of the network in the model plants *Arabidopsis thaliana* and *Oryza sativa*[12,19–21]. Similarly, phytoplankton in the ocean face remarkable environmental nutrient variations, and N and P nutrients are often limited[22,23]. Although the respective responses to N and P deficiencies have been extensively studied in phytoplankton[24,25], an integrative signaling pathway of N-P nutrition cross-talk has remained unknown until now. It is striking to find that trypsin, rather than homologs of plant *NRT1.1* and *NIGT1*[14,19], mediates and regulates the nitrate-phosphate signaling cross-talk.

The two-level model of *PtTryp2* function (Fig. 7f), including the direction of *PtTryp2* action and the direction of *PtTryp2* expression changes, demonstrate that *PtTryp2* functions by shifting the setpoints, by tuning its own expression level, at which N signaling or P signaling is triggered in response to environmental nutrient fluctuations so that cells commit to appropriate responses. However, much of the mechanics in the regulatory cascade, from environmental nutrient sensing, *PtTryp2*-mediated signaling, to the regulation of the effectors such as N- and Pi-transporters and assimilatory genes, remains to be elucidated. Although the interplay between N and P nutrition based on *SPX-NLP-NIGT1* and *SPX-PHR-NIGT1* cascades,

respectively have been uncovered in plants[12,19], how *PtTryp2* interacts with the *SPX-PHR* cascade[26] and whether a *SPX-NLP* cascade or other regulatory cascades exist and interact with *PtTryp2* for P and N nutrient regulation in phytoplankton remain to be addressed.

As an initial attempt, we have performed transcriptional regulatory interaction analysis based on the Inferelator algorithm[27] to predict the potential co-regulated genes in the *PtTryp2*-dependent regulatory cascade. Consequently, a set of 1034 genes co-regulated with *PtTryp2* were identified, including 10 transcription factors (Supplementary Table 4), 10 N metabolism and assimilation genes, and a P responsive gene (Supplementary Fig. 13). Moreover, the functional enrichment of the gene set showed that *PtTryp2* is possibly involved in post-transcriptional regulation, intracellular signal transduction pathway and kinase-based phosphorus metabolism and recycle pathway (Supplementary Fig. 14). The results hint on a potentially complex regulatory network that requires much more transcriptomes derived from more growth conditions than just the N and P conditions used in this study and other experimental approaches to unravel.

We used the potential co-regulated gene list identified in this study in a comparative analysis with the published co-regulatory analysis datasets that contained hundreds of public RNA-seq datasets: DiatomPortal[28] and PhaeoNet[29]. Interestingly, based on the DiatomPortal dataset, the *PtTryp2* was found in the Phatr_hclust_0381 hierarchical cluster that consists of 10 genes, which has been identified as the GO term of ubiquitin-dependent protein catabolism. In terrestrial plants, the ubiquitination and degradation of SPX4 was found to mediate the nitrate-phosphate interaction signaling pathway by enabling the release of PHR2 and NLP3 into the nucleus to activate the expression of both phosphate- and nitrate-responsive genes[12,19]. In addition, we found 120 genes that were common in our gene list and PhaeoNet, some of which are transcription factors.

Taken together, our analyses showed that the deletion and overexpression of *PtTryp2* simultaneously impacted nitrogen and phosphorus uptake, nitrogen and phosphorus contents of the cell, and the N:P ratio. The simultaneous impact on N and P in opposite directions suggests that this protein either directly regulates the N and P uptake machinery or is close to the direct regulator, e.g., functioning through the ubiquitination and degradation of the direct regulators as in terrestrial plants. Furthermore, it is conceivable that one or more intermediate relays between *PtTryp2* and the direct regulator would make it extremely challenging, if not impossible, to exert such precise and coordinated bidirectional regulation on N and P. To understand the mechanics of the regulatory mechanism, co-immunoprecipitation and Chromation immunoprecipitation sequencing are underway in our laboratory to experimentally identify the potential proteins and DNAs interacting with *PtTryp2*. Further studies on multiple fronts surrounding trypsin and its regulatory pathway are required for gaining an in-depth understanding of the interplay between N and P nutrition in phytoplankton and how phytoplankton will adapt to the potentially more variable and skewed N-P environment in the Anthropocene oceans.

## Methods

**Detection of trypsin genes in the *Tara* Oceans and PhyloDB databases**. An extensive search for putative trypsin and trypsin-like genes was performed in both the Marine Atlas of *Tara* Oceans Unigenes and eukaryotes metatranscriptomes (MATOUv1 + metaT)[30,31] and PhyloDB databases[32], using the BLASTP-algorithm based identification combined with hmmsearch analysis. The BLASTP-algorithm based search was conducted using trypsin amino acid sequences from NCBI and UniProt database as queries with an e-value ≤1e−5 as the threshold. For hmmsearch analysis, the trypsin (PF00089) and trypsin-like (PF13365) domain based profile hidden Markov models (HMM), with an E-value ≤1.0 E$^{-10}$. The

identified putative trypsin protein sequences were submitted to CDD (https://www.ncbi.nlm.nih.gov/Structure/bwrpsb/bwrpsb.cgi), Pfam and SMART (http://smart.embl-heidelberg.de/) to confirm the conserved trypsin domain. MATOU is a catalog of 116 million unigenes obtained from poly-A cDNA sequencing for samples of different size fractions from different water layers, which is available at the OGA website (http://tara-oceans.mio.osupytheas.fr/ocean-gene-atlas/). The geographic distribution of trypsin in phytoplankton based on *Tara* Oceans datasets were visualized using the maps R software package by RStudio (Version 1.4.1717). The PhyloDB is a custom-made database suitable for comprehensive annotation of metagenomics and metatranscriptomics data, comprised of peptides obtained from KEGG, GenBank, JGI, ENSEMBL, Marine Microbial Eukaryotic Transcriptome Sequencing Project, and various other repositories.

**Distance correlations between phytoplankton trypsin and environmental factors**. We computed pairwise distances between samples based on of phytoplankton trypsin relative abundances and the environmental nutrient factors from *Tara* Ocean datasets. The ambient nutrient conditions corresponding to trypsin expression data in the global ocean were extracted from PANGAEA and Ocean Gene Atlas[31,33]. The following nine environmental nutrient parameters were chosen for correlation analysis: iron_5m* (Fe_5m*, μmol/l), ammonium_5m* (NH$_4$_5m*, μmol/l), NO$_2$_NO$_3$ (μmol/l), NO$_2$ (μmol/l), NO$_3^-$ (μmol/l), PO$_4$ (μmol/l), Si (μmol/l), NO$_2$_5m* (μmol/l), NO$_3$_5m* (μmol/l). The nutrient factor labeled a star indicated the values estimated from oceanographic models. Based on our previous study, the phytoplankton trypsin from the 5–20 and 20–180 μm size fractions of surface water layer appeared to be more responsive to environmental stimuli, so the two size fractions data was selected for further analysis in this study. Given these distance matrices, we computed partial Mantel correlations between trypsin mRNA abundance and environmental data using the vegan R software package.

**P. tricornutum culture conditions**. The strain of *P. tricornutum* Bohlin used in this study (WT) was provided by the Center for Collections of Marine Algae, Xiamen University, China. As shown in a subsequent section, we generated *PtTryp2*-KO mutants and a blank vector control (KOC). Besides, we also generated a *PtTryp2*-overexpression (*PtTryp2*-OE or OE) transgenic clone and an OEC. The WT and the manipulated strains of *P. tricornutum* cells were generally grown in f/2 liquid medium[34] made with artificial seawater[35]. To create nitrate-depleted and phosphate-depleted conditions, f/2 medium was modified by reducing phosphate to 1 μM and omitting nitrate, respectively, with other nutrients remaining unchanged. The standard f/2, phosphate-replete and nitrate-replete cultures are named nutrient replete (HNHP), P-depletion (HNLP), and N-depletion (LNHP), respectively. Considering that the stock culture was kept in f/2 medium (882 μM NO$_3^-$ and 36.2 μM PO$_4^{3-}$), we pre-conditioned a pre-experiment master culture to N-depleted and P-depleted condition (pre-starvation). In this study, the pre-starvation treatment keeps the cellular N and P at a minimum level that cannot sustain cell growth. The starved culture was then used in subsequent experimental culture set up by providing varying nutrient combinations. Each condition being set up in triplicate. Cultures were grown at 20 °C in 16 h:8 h, light:dark diurnal cycles with a photon flux of 100 μE m$^{-2}$ s$^{-1}$.

**qRT-PCR to measure trypsin expressions in P. tricornutum**. To determine the growth stage expression levels of *PtTryp* genes in the WT strain, cells were harvested at the same time of the day (6 h after onset of the light period) on the 3rd, 5th, 7th, and 9th day. The time-course expression patterns of *PtTryp2* responded to nitrogen addition were analyzed based on public data[36]. The cells were harvested on 4 h after PO$_4^{3-}$ addition to explore the time-course expression patterns of *PtTryp2* to Pi supplement.

The cell cultures were sampled by centrifugation at 5000 × g, and the pellet was frozen immediately in liquid nitrogen. RNA was extracted using the TRIzol Reagent (Invitrogen, Carlsbad, CA, USA) and the RNeasy Plus Micro Kit (QIAGEN, Code: 74034, Germany), according to the previously reported method with minor modification[37]. RNA concentration was determined using a spectrophotometer (Nanodrop 2000: Thermo, USA). The cDNA template for qRT-PCR was synthesized with 1 μg of total RNA from each sample using PrimeScript®RT reagent Kit with gDNA Eraser (Perfect Real Time) (Takara, Code: DRR047A, Japan).

Primer Premier 5 was used to design primers specific to the *PtTryp* genes (Supplementary Table 5). qRT-PCR was carried out using the Sybr Green qPCR Kit Master Mix (2×) Universal (Bio-rad, US) on a Bio-Rad CFX96 Real-Time PCR System. Each reaction was in a 12 μl reaction volume containing 6.0 μl Sybr Green qPCR mixture, 1.0 μl of each 10 μM primer, 2.0 μl diluted cDNA (equivalent to 200 ng total RNA) and 2.0 μl ddH$_2$O. A two-step qRT-PCR program was employed: an initial denaturation step at 95 °C for 30 s followed by 40 cycles of 3 s at 95 °C and 30 s at 60 °C. The specificity of the qPCR products was assessed using melting curve analysis for all samples. Each culture condition had three biological replicates, each cDNA sample with three technical replicates. Following a previous study[38], the genes encoding hypoxanthine-guanine phosphoribosyltransferase (*PtHPRT*, Phatr3_J35566), TATA box binding protein (*PtTBP*, Phatr3_J10199), and ribosomal protein S1 (*PtRPS*, Phatr3_J10847) were used as the internal reference gene candidates. The web-based comprehensive tool RefFinder (http://blooge.cn/RefFinder/), which integrates currently available major computational program analysis (geNorm, Normfinder, BestKeeper, and the comparative Delta-Ct method) was used to assess the reference gene stability. Consequently, *PtTBP* is the most stable reference gene based on the web-based comprehensive tool RefFinder (Supplementary Table 6). Hence, the relative expression levels of the *PtTryp* genes were calculated using the $2^{-\Delta Ct}$ method by normalized to *PtTBP*.

**Generation of PtTryp2 knockout mutants**. To generate *PtTryp2* knockout mutants (*PtTryp2*-KO or KO) and corresponding control (KOC) using the *CRISPR/Ca9* gene editing technique, a transformation vector pKS-diaCas9-sgRNA with codon-optimized *Cas9* protein gene (*diaCas9*) and *P. tricornutum* U6 snRNA promoter[11] was constructed. The vector contained two *Bsa*I restriction sites located at the 5′-end of the guide RNA coding sequence, designed to facilitate the ligation of adapters (target sites) with 5′ TCGA and AAAC overhangs[39]. The *Cas9* target site of *PtTryp2* with the PAM signal (G-N19-NGG) was identified using Phyto-CRISP-Ex, a *CRISPR* target finding tool minimizing off-target potential[40]. The oligos that consisted of *Cas9* target sites (G-N19-NGG) and 5′ TCGA and AAAC overhang sequences were synthesized, then the adapter was made by annealing complementary oligos. The resulting adapter was ligated into the pKS-*diaCas9*-sgRNA plasmid using T4 DNA ligase (NEB). After confirming the sequence accuracy by sequencing, the pKS-*diaCas9*-sgRNA plasmid was introduced into *P. tricornutum* cells to induce mutation by the biolistic method. A pAF6 plasmid carrying Zeocin resistance gene (Invitrogen, Thermo Fisher Scientific, Grand Island, New York, USA) was designed and co-transformed to facilitate transformant selection[11]. A blank transformation control was constructed with the same procedure, but the vector only contained *diaCas9* and no *PtTryp2*-gRNA. All the primers used to generate the constructs are listed in Supplementary Table 5. All primers used in this study were synthesized by Sangon Biotech company (China).

To deliver the plasmids, *P. tricornutum* cells were collected from exponentially growing cultures and then concentrated to $2 \times 10^8$ cells ml$^{-1}$ at 3000 g for 5 min. Next, 200 μl of cells were spread on each 1.2% agar plate containing 50% seawater supplemented with f/2 medium. Tungsten M17 microcarriers were coated with vectors following the manufacturer's instructions (Bio-Rad). Transformation was performed using a Bio-Rad Biolistic PDS-1000/He Particle Delivery System (Bio-Rad, Hercules, California, USA) as described previously[41]. A burst pressure of 1550 psi and a vacuum of 28 Hg were used. The bombarded cells were transferred to selection plates (50 μg/ml Zeocin) 1 day after bombardment. After 3–4 weeks, the resistant colonies were transferred to liquid f/2 media (75 μg/ml Zeocin) to isolate the pure mutant strains.

**Generation of PtTryp2 overexpression transgenic lines**. The fusion protein of *PtTryp2* and green fluorescence protein (GFP) was designed to investigate phenotypic consequences of *PtTryp2*-overexpression (*PtTryp2*-OE or OE) and subcellular localization of *PtTryp2* in the *P. tricornutum* cells. To generate the fusion construct (pPha-T1-*PtTryp2*-eGFP), the CDS of *PtTryp2* was amplified and cloned into the pPhaT1 vector. These constructs were introduced into WT. In parallel, a blank vector was constructed to serve as an OEC. This construct (pPha-T1-*eGFP*) contained the CDS of *eGFP* but not that of *PtTryp2*. Both *PtTryp2*-OE and OEC vectors were sequenced to ensure sequence accuracy and then introduced into WT separately using the same biolistic procedure as described above. All the primers used to generate the constructs are listed in Supplementary Table 5. *PtTryp2* expression in the OE cell line was confirmed at a protein level through Western blot using anti-EGFP antibody. The anti-EGFP (Cat no. ab184601) and anti-GAPDH (Cat no. ab59164) antibodies used in this study were purchased from Abcam (England), and used after diluting them by 1000 and 10,000 times, respectively.

**Screening and genotype characterization for PtTryp2-knockout mutants**. Cell lysates of resistant colonies were prepared in lysis buffer (1% TritonX-100, 20 mM Tris–HCl pH 8, 2 mM EDTA) in an Eppendorf tube by repeated freezing and thawing. Five microliters of cell lysates were used for the PCR amplification of the genomic targets with specific primers compatible with sequencing (Supplementary Table 5). The PCR products were first analyzed by agarose gel electrophoresis. For checking the presence or absence of *Cas9* gene components in the genome, primer pairs were designed in the 3′ region for PCR to amplify the sequences (Supplementary Table 5). To confirm that the mutagenesis has caused frameshift insertions and deletions of *PtTryp2*, the target regions were amplified using specific primers designed flanking the targets (Supplementary Table 5). The PCR products were separated by electrophoresis on a 1% agarose gel, purified using MiniBEST Agarose Gel DNA Extraction Kit (Takara), and cloned into pMD19-T (Takara). Random clones were picked for Sanger sequencing.

**Subcellular localization by computational prediction and fusion protein assay**. Computationally, NUCPRED was used to predict the nuclear localization[42]. The presence of N-terminal signal peptides and transmembrane helices were predicted with SIGNALP 3.0, SIGNALP 4.1, TMHMM v.2.0, and PHOBIUS[43–46]. Several other programs, TARGETP, CHLOROP[47], PROTEOME ANALYST[48], EUK-MPLOC[49], MitoFates, HECTAR v.1.3[50], and ASAFIND v.1.1.5[51] were jointly used to predict possible localizations to the plastid, mitochondria, ER, and cytoplasm.

Experimentally, the *PtTryp2*-OE and OEC cells grown in standard f/2 media were used to investigate subcellular localization. The fluorescence signals were captured using the LSM780NLO confocal microscope (Carl Zeiss, Germany). Confocal images were collected and analyzed using the Zen software (ZENblue3-1_ZENblack_3-0SR-lite). Subcellular localization of *eGFP* was visualized with an excitation wavelength of 488 nm and emission wavelength of 510–540 nm. For subcellular locality reference, the nucleus was stained using Hoechst 33342 (Hoechst AG, Frankfurt, Germany) and visualized using an excitation wavelength of 346 nm and an emission bandpass of 430–600 nm. ER-Tracker Blue-White DPX (Invitrogen) was used to visualize ER with an excitation wavelength of 375 nm and emission bandpass of 550–640 nm. Golgi-Tracker Red (Beyotime, Beijing, China) was used to locate Golgi apparatus with an excitation wavelength of 589 nm and emission spectrum of 610–650 nm. The plastid autofluorescence was visualized with an excitation wavelength of 488 nm and an emission wavelength of 650–750 nm.

**Transcriptome profiling using RNA-seq.** The KOC and *PtTryp2*-KO1 cells were pre-starved in N- and P-depleted, and then inoculated into nutrient-replete (HNHP), N-depleted (LNHP), and P-depleted (HNLP) media as described above, each in triplicate. When the N- and P-depleted cultures started to show growth depression compared to the nutrient-replete cultures, cells were harvested. Total RNA was extracted as described above, and mRNA was purified using the Oligo(dT)-attached magnetic beads. The total RNA and isolated mRNA quality and quantity were checked using the Agilent 2100 Bioanalyzer and NanoDrop (Thermo Fisher Scientific, MA, USA). Libraries for RNAseq were created from 1 μg of mRNA from each culture. The resulting libraries were loaded into the patterned nanoarray, and single-end 50 bp reads were generated on BGIseq500 platform (BGI-Shenzhen, China), with a data output of about 22 M total clean reads for each library.

The sequencing data was filtered with SOAPnuke (v1.5.2)[52] by removing reads with sequencing adapter or low-quality base ratio and >5% unknown base ratio, and the resulting clean reads were stored in FASTQ format. The clean reads were mapped to the *P. tricornutum* reference genome (ftp.ensemblgenomes.org) using HISAT2 (v2.0.4)[53], and aligned to the reference coding gene set by Bowtie2 (v2.2.5)[54]. The expression level of a gene was calculated using RSEM (v1.2.12)[55]. Furthermore, DESeq2(v1.4.5)[56] was used to analyze differential gene expression with FoldChange ≥ 2 and *p* value ≤ 0.001 as the significance threshold. GO (http://www.geneontology.org/) and KEGG (https://www.kegg.jp/) enrichment analysis was performed by Phyper (https://en.wikipedia.org/wiki/Hypergeometric_distribution) based on Hypergeometric test. The significant levels of GO terms and pathways were corrected by *Q* value with a rigorous threshold (*Q* value ≤ 0.05) by Bonferroni[57].

**Measurement of cellular N and P contents.** The cellular N content was measured following the established protocol[58] with minor modification. In brief, 20 ml culture was filtered onto pre-combusted 25 mm GF/F filters, then dried in a 50 °C oven for over 12 h. The dry filters were then subjected to acid fumigation (1% hydrochloric acid) overnight at room temperature and dried again. Then, the dry filters were burned in the combustion tube, and the C and N were measured on PE2400 Series II CHNS/O elemental analyzer (PerkinElmer, USA) following the system's standard protocol.

For cellular phosphorus content measurement, 20 ml culture was filtered onto pre-combusted a 25 mm GF/F filter. The cells were digested to release phosphorus as orthophosphate following the persulfate oxidation technique, and the concentration of phosphate was then measured using the ascorbic acid method by spectrophotometer.

**Determination of NO₃⁻ and Pi uptake rates.** To determine the nutrient consumption of algal cells from medium, concentration changes for phosphate and nitrate over time were determined. Based on previous experiments, 36.2 μM $PO_4^-$ was exhausted from the f/2 media by ~$5 \times 10^5$ cells ml$^{-1}$ in 24 h, when $NO_3^-$ remained detectable. Moreover, the cultures were grown in 16 h:8 h, light:dark cycles. Based on this set of information, the concentration changes for phosphate and nitrate were determined at 16 h (experience 8 h each for light and dark periods) after addition. A 20 ml culture sample was collected from each triplicate culture under each growth condition, and serially filtered through 25 mm GF/F membrane and 0.22 μm sterile membrane. The final filtrate was saved and stored at −20 °C until nutrient measurements. One ml subsample was removed for cell count. Nitrate concentrations were measured using the chromotropic acid method. Phosphate concentrations were measured using the ascorbic acid method. Both measurements were carried out using a spectrophotometer (V-5600, METASH, China) and commercial reagent sets. The percentage nutrient removal and nutrient uptake rate per cell were calculated from the changes of the nutrients over time and cell concentrations following Aigars et al.[59].

**Gene co-expression network analysis.** The integrative transcriptomic data comprise 94 samples across different nitrogen and phosphorus conditions were retrieved from this study and published studies[26,36,60]. The expression data were analyzed by the Inferelator algorithm[27] to identify regulatory networks that could emerge from *PtTryp2*. The Inferelator algorithm generates a confidence score of ranging from 0–1 where 1 is near certainty of a regulatory relationship. Here, we deployed a cutoff of 0.70 to obtain high quality correlations with *PtTryp2*. Once the regulatory network was generated, we performed GO-term enrichment analysis using the latest annotations available for *P. tricornutum* to assess what functions could be regulated by trypsin.

**Statistics and reproducibility.** All statistical tests used are described in figure legends. All *n* values indicated in the figures represent independent experimental samples and not technical replicates. Results are expressed as the mean ± standard deviation. The comparisons between the averages of the two groups were evaluated using the one-tailed Student's *t* test using GraphPad Prism software (version 9). *p* values of ≤0.05 were considered statistically significant. The exact *p* values with significance are shown in the figures.

**Reporting summary.** Further information on research design is available in the Nature Research Reporting Summary linked to this article.

## Data availability
The RNA-seq datasets are available at NCBI in the GEO (Gene Expression Omnibus) database under accession number GSE202896. Publicly-available trypsin sequences were obtained from Pfam (https://pfam.xfam.org/family/PF00089 and https://pfam.xfam.org/family/PF13365), and Ensemble databases (http://protists.ensembl.org/Phaeodactylum_tricornutum/Info/Index) Worldwide distributions and abundances of trypsin genes were acquired from the publicly-available OGA datasets (https://tara-oceans.mio.osupytheas.fr/ocean-gene-atlas/). All other data that support the findings of this study are available from the corresponding authors upon reasonable request. Source data are provided with this paper.

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

## Acknowledgements

We wish to thank Dr. Kaidian Zhang, Dr. Liying Yu, Dr. Xin Lin, Ms. Chentao Guo, Mr. Chenmin Zhu, and Mr. Yujie Wang for technical and logistic assistance. We also wish to thank the Beijing Genomics Institute (China) for RNA-seq services. This work was supported by the National Natural Sciences Foundation of China (grant # 41906123 to Y.Y.), the Open Fund of CAS Key Laboratory of Marine Ecology and Environmental Sciences, Institute of Oceanology, Chinese Academy of Sciences (grant # KLMEES202006 to Y.Y.), the Marine S&T Fund of Shandong Province for Pilot National Laboratory for Marine Science and Technology (Qingdao) (grant # 2018SDKJ0406-3 to S.L.), the MEL Internal Programs of the State Key Laboratory of Marine Environmental Science, Xiamen University (grant # MELRI2105 to S.L.). Lin was in part supported by the Gordon and Betty Moore Foundation (GBMF grant #4980.01 to S.L.).

## Author contributions

S.L. and Y.Y. conceived and designed the research; Y.Y., X.S., M.M., J.H., and L.L. performed the experiments; S.L., Y.Y., X.S., and F.W.P. did the data analysis; S.L. and Y.Y. wrote the manuscript.

## Competing interests

The authors declare no competing interests.
