## [Peer Review File · Nature Communications]

Trypsin as a coordinate regulator of N and P nutrients in marine phytoplanktonReviewers' comments:

Reviewer #1 (Remarks to the Author):

This is an interesting study, performing a holistic analysis of trypsin functions in maintaining cellular N: P homeostasis in *Phaeodactylum*. The limited number of mutants, generated from one CRISPR region, unfortunately makes it difficult to attribute the phenotypes observed specifically to the trypsin targeted, despite the very thorough physiological validation, and I would encourage the authors to resubmit the paper once they have either generated additional mutants, or tested complementation of their existing lines, as a new submission. I provide more detailed comments below:

Fig. 1A, B: I do not see the point in confirming purely the presence of trypsin, which is presumably a very common gene, across all stations in Tara Oceans, or across the algal ToL (e.g., given there are ten homologues in *Phaeodactylum*)- I would expect to see it everywhere.

Good questions that could be asked instead: for the Tara analyses, in which stations are trypsin genes most abundant, and how does this correlate to measured environmental parameters such as N and P concentrations? The authors should test this for each depth and size combination fraction within the metaT data, and ideally normalise against something like metaG total abundances to differentiate between stations with lots of algae and lots of trypsins.

As a general note, an evalue of 1E-05 is really too weak to conclude that a match is indeed a trypsin meta-gene or something structurally similar but functionally different. I would suggest the authors verify the identity of the trypsin meta-genes selected either by the presence of conserved domains or by phylogeny, and calculate abundances purely from the meta-genes that unambiguously encode trypsin proteins.

For the distribution of trypsin in cultured algae, perhaps the authors could describe a bit more its evolutionary history, how different homologues from different algal groups relate to one another; whether there are different trypsin subfamilies, duplications or horizontal transfers across the tree of life. A particularly pertinent question would be if higher copy numbers of trypsin genes are associated with algae sampled from N-rich and P-limited environments, which would help confirm any inferences from Tara data that the physiological function of PtTryp2 is conserved in other algae.

Fig. 1E: not really convinced by the localisation of PtTryp2, largely due to the limited resolution and possible ER-Tracker overstraining of the images. I suggest reimaging the cell lines with a higher resolution objective or greater line/ frame averaging to get a clear picture, ideally presenting a few different Z-sections of the same cell to confirm the 3D distribution of each signal (or at least present a bright-field photo alongside so we can be sure the cells are in focus!), and present stain-negative and GFP-negative control images to exclude the possibility of crosstalk between the ER and GFP channels.

Fig. 1F. One of the PtTryp2-KO lines has an 18 bp (in-frame) deletion- can it really therefore be said to have a loss of PtTryp2 function? This would need to be verified another way, e.g. qPCR to identify diminished PtTryp2 expression, or enzymatic assay of each individual mutant line. If in doubt, exclude it from the subsequent physiological calculations.

As a general rule, phenotypes observed from mutants generated from purely one CRISPR region, even if multiple mutants are generated with consistent phenotypes, may be the result of secondary mutations due to off-target activity of the CRISPR sequence. This is particularly an issue for mutants generated through biolistic transformation due to permanent integration of the CAS9 sequence into the recipient genome, dramatically increasing the probability of secondary mutations, even if the authors have selected a CRISPR sequence with minimal off-target potential via phyto-CRISPEX. Ideally, the authors need either at least one other PtTryp2 KO line targeting a different CRISPR region with the same phenotype, or to complement their mutants with a wild-type copy of the PtTryp2 gene, with a suppression of the mutant phenotype, to be certain their phenotype is exclusively due to reduced ptTryp2 activity.

Figs. 2, 3- I would hesitate to over interpret the number of DEGs observed in the ptTryp2-KO lines as being significantly different to controls, as this depends also on the variance (i.e., size of the standard deviations) of RNAseq samples from each gene. Can the authors look into this e.g. considering the relative abundances of housekeeping gene transcripts in each sequence library? The authors also need to confirm the number of biological replicates performed for each RNAseq experiment, to be sure that the DEGs at large are reproducible between samples.

Fig. S3B. The overexpressor lines appear to be triradiate, which is likely to substantially change gene expression and physiological patterns in itself (c.f. Rastogi et al. 2018, Zhao et al. 2020, Galas et al. 2021). What morphology were the OEC cultures used for physiological analysis? If a stable (GFP-expressing) fusiform line has been established since, I recommend repeating the N and P uptake experiments with this line.

Fig. S4. Actually looking at this the Hoechst and Golgi-tracker stains are better, I suspect that the ER-tracker is just overstained- suggest repeating this with a lower stain concentration or shorter incubation time, alongside providing control images, of course.

Lines 74-75: should be "in the chloroplast via the secretory pathway"

Line 76: is it really true to say that N and P assimilation occur in the chloroplast? Agreed that this is where they are incorporated into organic compounds (I.e. via the glutamate/ oxoglutarate cycle and phosphorylation of ADP, respectively), although this should be clarified and supported by relevant citations. NB of course that parallel pathways for both of these processes occur in the diatom mitochondria (c.f. Smith et al. 2019): it would be interesting to know accordingly MitoFates and HECTAR predictions for other Phaeodactylum trypsins in Table S2, to verify if any encode mitochondria-targeted proteins.

Lines 79, 80: if ptTryp2 is indeed localised to both the cER and plastid (as described above), it should also logically localise to the PPC. The authors could test these hypotheses explicitly using self-assembling GFP constructs, using known PPC (Hsp70) and cER (BiP) reporters, alongside perhaps a cytoplasmic and pyrenoid negative control.

Lines 108-120: given the centrality of ptTryp2 expression patterns to this paper, the qPCR verification of this absolutely needs to be shown in a main text figure rather than fig. S2. I do not see a significant repression of PtTryp2 expression in N- from fig. S2C, but perhaps this is a scale issue (plot on a log scale to expose differences between two low levels of expression?), and Pvalues should of course be shown.

Lines 127 onwards: suggest not using the acronym PSI, as this could also refer to Photosystem I (and was misread by me on my initial scan of the paper)

Lines 145-177: to uncover possible mediators of PtTryp2 regulation of N: P homeostasis, the authors could consider looking for highly coregulated Phaeodactylum genes to PtTryp2 in published meta-studies of Phaeodactylum gene expression (e.g. Ashworth et al. 2016; Ait-Mohamed et al. 2020). These could reveal possible signaling partners or cofactors (which may show positive transcriptional coregulation trends to PtTryp2) or indeed substrates of PtTryp2 activity (which may show strong negative relationships). It would be helpful also if the authors could at least discuss future experimental approaches (e.g., yeast two-hybrid assays, BioID) that could be used to unravel possible molecular functions of PtTryp2.

Lines 375- 378: what time of day were the cells samples, was this kept consistent to avoid variant Circadian effects on gene expression?

Lines 489-494 and Fig. S5: what localisations were identified for the proteins encoded by DEGs? Was there any inferred bias towards chloroplast-targeted proteins (given the inferred localisation of ptTryp2), secretory/ endomembrane proteins (which may be consistent with changes to N and P uptake) or mitochondrial proteins (N recycling, particularly in the context of the ornithine/ urea cycle).

Fig. S6- plot Pvalues.

Supporting dataset- looks clear enough, although a contents page would be helpful.

Reviewer #2 (Remarks to the Author):

Overview

In this manuscript the authors demonstrate that trypsin is found in many phytoplankton genomes. They identify that a trypsin orthologue in the diatom *Phaeodactylum* (PtTryp2) plays a role in regulating the N:P ratio. This is a significant discovery as little is currently known about the mechanisms that enable phytoplankton are able to alter their stoichiometry under changing nutrient availability. Overall the work is highly novel and the finding that trypsin is involved with the coordinating N and P metabolism is intriguing and has the potential to substantially advance this field of research.

However, I'm not convinced that the term 'master regulator' should be applied to trypsin as its mode of action in this scenario remains unknown. I also think that some of the conclusions of the authors are not fully justified by their data and that some further experimental details need to be provided as detailed below.

Specific comments

Title and line 30. Trypsin is a master regulator. The authors provide convincing evidence that knockout or overexpression of Tryp2 alters the N:P ratio of *Phaeodactylum*. Trypsin is a protease that plays an important role in the breakdown of proteins in human nutrition. It is therefore not clear how the activity of trypsin enables it to coordinate different aspects of the N/P metabolism. Whilst the authors freely acknowledge this, I think the lack of a mechanism precludes the description of Tryp2 as a master regulator. This term is normally used to proteins such as transcription factors that directly control the activity of all the downstream proteins in a specific pathway – this has not been demonstrated.

Line 65 Tryp2 expression is downregulated under N-limiting conditions. Whilst Tryp2 is clearly upregulated under P limitation, the down regulation under N-limitation is not clear to me. To display up- and down-regulation equally, it is better to show gene expression as log fold change (as in used for transcriptomic data in Fig 3a). In Fig 2c Tryp2 expression increases substantially from day 3 to day 5 in N-limited cells (and N-replete control cells), but presumably N will have depleted substantially from day 3 to day 5. I agree that Tryp2 expression in N-limited cells is lower than the control at day 9, but it is still higher than N-limited cells at day 3. More details need to be provided in the figure legend of Fig1c to explain this. In the Methods it states Tryp2 expression was normalised to three reference genes, but in Extended Data Fig 2 expression is shown normalised to a single reference gene (PtTBP). The data in Fig2C is presumably shown relative to expression of Tryp2 in control (N+P+) cells on day 3 but this all needs to be described clearly. Replication and errors bars also need to be described.

As the authors subsequently propose that Tryp2 controls N/P ratio, it would interesting to show how its expression co-varies with N/P ratio.

Line 91 trypt2 mutants. Three mutants were generated and are all described as frameshift mutants (line 287), but one deletion (18 bp) doesn't appear to introduce a frameshift. Only one mutant was used for all subsequent studies, but it is not clear which one was used (line 96). Was the phenotype of the other mutants investigated? The effect on N and P acquisition would be more convincing if demonstrated in multiple mutants, although the authors are able to show that over-expression provides the converse phenotype. The authors do not show growth data for the mutants. It would be interesting to see if they have a defect in growth under N or P limitation.

Line 93 Upregulation of N assimilation. Inactivation of Tryp2 led to upregulation of most N

assimilation genes (Fig 2a). Whilst many N assimilation genes are upregulated, the pattern of gene expression is very different from N-limited wild type cells, i.e. the genes important in N-limitation are not upregulated in try2 KO – this should be made clear

Line 129 try2 knockout downregulates PSI genes but upregulates SPX genes. This effect isn't very clear in Fig 4a (KO vs KOC in nutrient replete cells).

Line 143 Tryp2 activates PSI genes. Without a clear indication of mechanism, I don't know the authors should state that Tryp2 functions to activate PSI genes. This infers a direct mode of action of Tryp2 on these genes that has not been demonstrated.

Line 165 Abolishment of Pi uptake repression. There is very little repression of Pi uptake shown in the wild type (KOC), with very similar rates of Pi uptake in N-replete versus N-limited cells.

Fig 3 and 4. When the cells were harvested for transcriptomic analysis is not made clear. In the Methods, the sentence beginning line 366 does not make sense ('Cells were pre-starvation following the steps:'). Describing the nutrient status of the cells is really important for the interpretation of the transcriptomic data.

Minor points

Line 44 The term Plantae is usually used to refer to the Archaeplastida. Most phytoplankton do not belong to the group.

Line 82, Figure 1. The localization of Tryp2 in the chloroplast is not particularly clear in the figure. Is a transmitted light image also available? It would help to show the cytosolic-localised GFP control (Extended data 4).

Fig 1c and throughout. The labels N-P+ need be much clearer. Showing N- or N+ in superscript is very difficult to distinguish in the figures.

Fig 3a and 4a – N and P assimilation genes in these figures should be labelled with identifiers (as in Extended data 5).

Responses to the comments from reviewers

Reviewer #1:

Comment 1: This is an interesting study, performing a holistic analysis of trypsin functions in maintaining cellular N: P homeostasis in *Phaeodactylum*. The limited number of mutants, generated from one CRISPR region, unfortunately, makes it difficult to attribute the phenotypes observed specifically to the trypsin targeted, despite the very thorough physiological validation, and I would encourage the authors to resubmit the paper once they have either generated additional mutants or tested complementation of their existing lines, as a new submission. I provide more detailed comments below.

Response 1: Thank you for your constructive comments and suggestions. We have revised the manuscript as your suggestions. In the revised revision, we have done more experimental work and clarification to support our claims. We added two additional pure KO mutants, one with 8bp deletion and the other with 41bp deletion. The transcriptomic and physiological experimental results have been achieved for 6 mutants (KOC, KO1, KO2, KO3, OEC and OE) and presented in the revised manuscript. These results were consistent with our original observations in the initial submission, verifying that KO cell lines indeed are loss-of-function mutants of *PtTryp2* and OE enhances function of *PtTryp2*, and as such they can be used for subsequent functional analysis of *PtTryp2*.

As shown in Figure 4c, compared with control cell lines (KOC and OEC), all of the three *PtTryp2*-KO lines with a significantly diminished *PtTryp2* expression under both nutrient depletion and repletion, while the OE cell line showed in converse. In addition, the expression of OE cell line was confirmed at protein level by Western blot (Fig. 3a). Moreover, the *PtTryp2* expression level in KOC cell lines can dramatically influence response to the ambient N and P fluctuation, but consistently showed a constant low

expression pattern in KO lines, indicating that the function of *PtTryp2* was really abolished in KO lines (Fig. 4d). Moreover, the cellular N/P ratio was significantly increased by the inactivation of *PtTryp2*, but significantly decreased by the overexpression of *PtTryp2* (Fig. 7d and 7e). In addition, we have carried out the gene expression pattern analysis of *PtTryp2* from WT across different N/P ratio cultivation conditions, indicating that *PtTryp2* showed expression co-varies with the N/P ratio (Fig. 7c). Moreover, the time-course analysis showed that *PtTryp2*'s expression fluctuated less at an N/P ratio of 16:1 compared to other N/P ratios. The N/P ratio of 16:1 has been proven as optimal for *P. tricornutum* growth (Extended Data Fig. 11). All our data demonstrate in concert that *PtTryp2* is a coordinate regulator of N-P stoichiometric homeostasis.

Comment 2: Fig. 1A, B: I do not see the point in confirming purely the presence of trypsin, which is presumably a very common gene, across all stations in Tara Oceans, or across the algal ToL (e.g., given there are ten homologues in *Phaeodactylum*)- I would expect to see it everywhere. Good questions that could be asked instead: for the Tara analyses, in which stations are trypsin genes most abundant, and how does this correlate to measured environmental parameters such as N and P concentrations? The authors should test this for each depth and size combination fraction within the metaT data, and ideally normalize against something like metaG total abundances to differentiate between stations with lots of algae and lots of trypsins.

Response 2: Thanks for this great idea. We indeed had performed the analysis and included the result in another paper entitled “An ancient enzyme finds a new home: prevalence and neofunctionalization of trypsin in marine phytoplankton” (submitted to Research Square as preprint: DOI: 10.21203/rs.3.rs-1008417/v1). Hence, in the revised manuscript, we cited that paper and discussed the finding, whilst for the present manuscript we additionally analyzed distance-corrected dissimilarities of

phytoplankton trypsin transcript abundance with environmental nutrient factors using the partial Mantel test, and identified environmental N and P as the most correlated factors to modulate the expression of trypsin. The findings have been added to the revised manuscript.

Comment 3: As a general note, an e-value of 1E-05 is really too weak to conclude that a match is indeed a trypsin meta-gene or something structurally similar but functionally different. I would suggest the authors verify the identity of the trypsin meta-genes selected either by the presence of conserved domains or by phylogeny, and calculate abundances purely from the meta-genes that unambiguously encode trypsin proteins.

Response 3: We agree with this suggestion, and we actually have done that. We identified the trypsin and trypsin-like candidate genes taking the following steps: Firstly, a BLASTP-algorithm based search was conducted using trypsin amino acid sequences from NCBI and UniProt database as queries with an e-value $\leq 1e-5$ as the threshold. Secondly, the hmmsearch analysis was conducted. We downloaded the HMM profile of trypsin and trypsin-like (PF00089 and PF13365) from Pfam protein family database ([http:// pfam.xfam.org/](http://pfam.xfam.org/)) and used it as the query ($P < 0.001$) by hmmsearch from the target hits. Thirdly, the identified putative trypsin protein sequences were submitted to CDD ([https:// www.ncbi.nlm.nih.gov/Structure/bwrpsb/bwrpsb.cgi](https://www.ncbi.nlm.nih.gov/Structure/bwrpsb/bwrpsb.cgi)), Pfam and SMART (<http://smart.embl-heidelberg.de/>) to confirm the conserved trypsin domain. We did not clearly describe it in the original version and have revised it in the new version.

Comment 4: For the distribution of trypsin in cultured algae, perhaps the authors could describe a bit more its evolutionary history, how different homologues from different algal groups relate to one another; whether there are different trypsin subfamilies, duplications or horizontal transfers across the tree of life. A particularly pertinent

question would be if higher copy numbers of trypsin genes are associated with algae sampled from N-rich and P-limited environments, which would help confirm any inferences from Tara data that the physiological function of *PtTryp2* is conserved in other algae.

Response 4: Another excellent idea, and again in the other manuscript (preprint) mentioned above, we have done a comparative genomic-wide analysis of trypsin genes based on available sequenced genomes that span all the major algal phyla. By systematic comparative analyses, we documented lineage-specific diversity and expansion of trypsin genes in the evolution of marine phytoplankton. Genome-wide analyses showed that trypsin genes were more extensively duplicated in diatoms than other lineages. All identified trypsin genes were clearly classified into the classical trypsin and divergent variants (trypsin-like) clades, and with subfamily-specific motif and domain distribution patterns. The distribution of trypsin clusters on chromosomes, suggests the tandem duplication contributes to the expansion of this family in diatom. Moreover, the expression of trypsin genes could be modulated by different environmental factors, and while their expression in dinoflagellates appears to be constitutive, that in diatoms is particularly responsive to environmental stimuli. We posit that trypsin genes have important functions in phytoplankton, and that the duplication and neofunctionalization of trypsin genes may be important in diatoms to adapt to dynamical environmental conditions, contributing to diatoms' dominance in the coastal oceans. Considering that in this study we mainly focused on unraveling the function of trypsin, we did not include that part but cited it in this manuscript and still feel that it would complicate the manuscript if we move all that material here.

Comment 5: Fig. 1E: not really convinced by the localization of *PtTryp2*, largely due to the limited resolution and possible ER-Tracker overstraining of the images. I suggest reimaging the cell lines with a higher resolution objective or greater line/ frame

averaging to get a clear picture, ideally presenting a few different Z-sections of the same cell to confirm the 3D distribution of each signal (or at least present a bright-field photo alongside so we can be sure the cells are in focus!), and present stain-negative and GFP-negative control images to exclude the possibility of crosstalk between the ER and GFP channels.

Response 5: As suggested, we reimaged the cell lines that were stained with different concentration dyes, different staining times, and different negative controls. As shown in Fig. 3, Extended Data Figs.7 and 8, the results of stain-negative and eGFP-negative control images exhibited no co-localization, indicating that there was no interference between the ER-Tracker (Ex=375 nm, Em=550-640 nm) and eGFP (Ex=488 nm, Em=510-540 nm) channels. We also tried the Z-stack imaging but were unable to obtain higher resolution images due to the thinness of *P. tricornutum* cells. However, we have a bright-field photo (transmission light channel) in the figures, which provides a better cell compartment context. Compared with the images of OE cell lines, the signal of eGFP in OEC was distributed randomly in the cytoplasm, but did not show co-localization with chloroplast and ER.

Comment 6: Fig. 1F. One of the PtTryp2-KO lines has an 18 bp (in-frame) deletion- can it really therefore be said to have a loss of PtTryp2 function? This would need to be verified another way, e.g. qPCR to identify diminished PtTryp2 expression, or enzymatic assay of each individual mutant line. If in doubt, exclude it from the subsequent physiological calculations.

Response 6: Thanks for catching that. Indeed, 18b deletion is not frameshift and we can not say for sure it causes loss of function. During the revision, we obtained more mutant clones, which are all frameshift mutations. We have added the qPCR data of KO vs. KOC, OE vs. OEC lines in Fig. 4c, it showed that all of the three PtTryp2-KO lines with a significantly diminished *PtTryp2* expression, while a significant increase

expression was observed in the OE line. All of these results verify that the PtTryp2-KO lines have a loss of *PtTryp2* function, and OE has a magnified *PtTryp2* function. Moreover, the *PtTryp2* expression level in KOC cell lines can influence dramatically response to the ambient N and P level, but consistently showed a constant low expression pattern in KO lines, indicating that the function of *PtTryp2* was really abolished in KO lines (Fig. 4d).

Comment 7: As a general rule, phenotypes observed from mutants generated from purely one CRISPR region, even if multiple mutants are generated with consistent phenotypes, may be the result of secondary mutations due to off-target activity of the CRISPR sequence. This is particularly an issue for mutants generated through biolistic transformation due to permanent integration of the CAS9 sequence into the recipient genome, dramatically increasing the probability of secondary mutations, even if the authors have selected a CRISPR sequence with minimal off-target potential via phyto-CRISPEX. Ideally, the authors need either at least one other PtTryp2 KO line targeting a different CRISPR region with the same phenotype, or to complement their mutants with a wild-type copy of the PtTryp2 gene, with a suppression of the mutant phenotype, to be certain their phenotype is exclusively due to reduced PtTryp2 activity.

Response 7: We agree with your opinions and suggestions, and we have verified that the Cas9 gene was still existed in the PtTryp2-KO lines using PCR. Considering precisely this possibility you mentioned, we compared the physiological verification of multiple KO mutants with overexpression lines, and found that most of the physiological parameters of the KO mutants were completely opposite to overexpression lines (Figs. 5, 6). In addition, the gene expression level of *PtTryp2* was found significantly decreased in KO lines, and did not respond to ambient nitrogen and phosphorus varying (Fig. 4). Taken together, these results demonstrate that the PtTryp2-KO lines with a loss of function of PtTryp2.

Comment 8: Figs. 2, 3- I would hesitate to over interpret the number of DEGs observed in the PtTryp2-KO lines as being significantly different to controls, as this depends also on the variance (i.e., size of the standard deviations) of RNAseq samples from each gene. Can the authors look into this e.g. considering the relative abundances of housekeeping gene transcripts in each sequence library? The authors also need to confirm the number of biological replicates performed for each RNAseq experiment, to be sure that the DEGs at large are reproducible between samples.

Response 8: This is a great idea. As you suggested, we conducted sample relationship analysis on biological replicates based on 11 classic housekeeping gene transcripts to verify the reproducibility of findings. As shown in Extended Data Fig. 9, the correlation analysis based on the selected housekeeping gene transcripts was very high (> 0.94), indicating that there was a good reproducibility between samples. The 11 putative housekeeping gene lists were retrieved according to Siau et al.'s research (Siau et al., 2007). Besides, as described in the Methods section, the DEGs were analyzed using DESeq2 with $\text{foldchange} \geq 2$, $\text{P-value} \leq 0.001$ as the significance threshold. In this work, three biological replicates were performed for each RNAseq experiment.

Comment 9: Fig. S3B. The overexpression lines appear to be triradiate, which is likely to substantially change gene expression and physiological patterns in itself (c.f. Rastogi et al. 2018, Zhao et al. 2020, Galas et al. 2021). What morphology were the OEC cultures used for physiological analysis? If a stable (GFP-expressing) fusiform line has been established since, I recommend repeating the N and P uptake experiments with this line.

Response 9: We agreed with you that different morphologies are likely to substantially change gene expression and physiological patterns in themselves. All the seven cell lines (Pt-WT, KOC, KO1, KO2, KO3, OEC, and OE) used in this work have the same morphological profile, which contains both triradiate and fusiform. Because we did not

notice any difference in the proportion of each of the two morphotypes, we think that the results should reflect the genetic differences rather than morphological differences.

Comment 10: Fig. S4. Actually, looking at this the Hoechst and Golgi-tracker stains are better, I suspect that the ER-tracker is just overstained- suggest repeating this with a lower stain concentration or shorter incubation time, alongside providing control images, of course.

Response 10: As you suggested, we have reimaged the cells at different stain concentrations, different staining times and different negative controls (Extended Data Figs. 7, 8). The results remain similar at those different concentrations.

Comment 11: Lines 74-75: should be "in the chloroplast via the secretory pathway"

Response 11: We have revised the text as you suggested.

Comment 12: Line 76: is it really true to say that N and P assimilation occur in the chloroplast? Agreed that this is where they are incorporated into organic compounds (I.e. via the glutamate/ oxoglutarate cycle and phosphorylation of ADP, respectively), although this should be clarified and supported by relevant citations. NB of course that parallel pathways for both of these processes occur in the diatom mitochondria (c.f. Smith et al. 2019): it would be interesting to know accordingly MitoFates and HECTAR predictions for other *Phaeodactylum* trypsins in Table S2, to verify if any encode mitochondria-targeted proteins.

Response 12: As you suggested, we have modified terminology throughout the text as appropriate and added the relevant citations. Moreover, we used the MitoFates and HECTAR to predict the subcellular location of identified trypsins, the results have been added in Table S2, but no trypsin was predicted to locate in mitochondria.

Comment 13: Lines 79, 80: if *PtTryp2* is indeed localized to both the cER and plastid (as described above), it should also logically localize to the PPC. The authors could test these hypotheses explicitly using self-assembling GFP constructs, using known PPC (Hsp70) and cER (BiP) reporters, alongside perhaps a cytoplasmic and pyrenoid negative control.

Response 13: Thanks for the suggestion, which sounds like a great idea for future continued study. For this manuscript, we would like to focus more on the function, and can confidently say that *PtTryp2* is localized in the chloroplast, and probably is transported to the chloroplast via the ER.

Comment 14: Lines 108-120: given the centrality of *PtTryp2* expression patterns to this paper, the qPCR verification of this absolutely needs to be shown in the main text figure rather than fig. S2. I do not see a significant repression of *PtTryp2* expression in N- from fig. S2C, but perhaps this is a scale issue (plot on a log scale to expose differences between two low levels of expression?), and P-values should of course be shown.

Response 14: As you suggested, we have placed the results of *PtTryp2* gene expression pattern in the main text figure (Fig. 2), and changed the Y axis scale and shown P values to make the figures clearer.

Comment 15: Lines 127 onwards: suggest not using the acronym PSI, as this could also refer to Photosystem I (and was misread by me on my initial scan of the paper)

Response 15: We have revised it as you suggested, the "PSI" was corrected as "P starvation-induced".

Comment 16: Lines 145-177: to uncover possible mediators of *PtTryp2* regulation of N: P homeostasis, the authors could consider looking for highly coregulated *Phaeodactylum* genes to *PtTryp2* in published meta-studies of *Phaeodactylum* gene

expression (e.g. Ashworth et al. 2016; Ait-Mohamed et al. 2020). These could reveal possible signaling partners or cofactors (which may show positive transcriptional coregulation trends to PtTryp2) or indeed substrates of PtTryp2 activity (which may show strong negative relationships). It would be helpful also if the authors could at least discuss future experimental approaches (e.g., yeast two-hybrid assays, BioID) that could be used to unravel possible molecular functions of PtTryp2.

Response 16: Thanks for the excellent suggestions. We used the Inferelator algorithm (Bonneau R, *et al*, 2006) to predict the co-regulated genes to PtTryp2 based on our transcriptomic data. The integrative transcriptomic data from 94 samples encompassing different nitrogen and phosphorus conditions were retrieved from this study and recent published studies comprise (Smith et al. 2019, Zhang et al. 2021, Li et al. 2022). We found a set of 1034 genes that appeared to be co-regulated with PtTryp2 at a confidence interval of 70%. A further examination of the gene set revealed a regulatory relationship between PtTryp2 and several nitrogen assimilation and metabolism genes, a P responsive gene, and ten transcription factors (Extended Data Fig. 12). Moreover, the functional enrichment of the gene set showed that PtTryp2 possibly regulates post-transcriptional regulation pathway, intracellular signal transduction pathway and a set of kinases related to phosphorus metabolism and recycle pathway (Extended Data Fig. 13). In addition, we also mention in the manuscript that we are conducting Co-immunoprecipitation (co-IP) and Chromatin immunoprecipitation sequencing (ChIP-seq) in our laboratory as part of the required research effort to experimentally identify the proteins and DNA that PtTryp2 interacts with. Yeast two-hybrid assays and BioID will be explored as well.

Comment 17: Lines 375- 378: what time of day were the cells samples, was this kept consistent to avoid variant Circadian effects on gene expression?

Response 17: All the timepoint of cells sampling were kept consistent, at the 6 hours after onset of the light period. We have clarified the cell sampling in the new version.

Comment 18: Lines 489-494 and Fig. S5: what localizations were identified for the proteins encoded by DEGs? Was there any inferred bias towards chloroplast-targeted proteins (given the inferred localization of PtTryp2), secretory/ endomembrane proteins (which may be consistent with changes to N and P uptake) or mitochondrial proteins (N recycling, particularly in the context of the ornithine/ urea cycle).

Response 18: Following your suggestion, we have used the MitoFates and HECTAR to predict the localization of identified DEGs, and placed the results in Supplement Tables 2 and 3. Except for transporters, most of the other DEGs related to nitrogen assimilation and metabolism showed biases towards chloroplast- and mitochondrion-targeting. However, most of the phosphorus responsive genes were predicted to target other localizations, but with a signal peptide.

Comment 19: Fig. S6- plot Pvalues.

Response 19: We have rearranged the figures, and plotted P values.

Comment 20: Supporting dataset- looks clear enough, although a contents page would be helpful.

Response 20: We have made the Extended Data figure legends clearer in the revised manuscript.

Reviewer #2:

Comment 1: In this manuscript, the authors demonstrate that trypsin is found in many phytoplankton genomes. They identify that a trypsin orthologue in the diatom *Phaeodactylum* (PtTryp2) plays a role in regulating the N:P ratio. This is a significant discovery as little is currently known about the mechanisms that enable phytoplankton are able to alter their stoichiometry under changing nutrient availability. Overall, the work is highly novel and the finding that trypsin is involved with the coordinating N and P metabolism is intriguing and has the potential to substantially advance this field of research.

However, I'm not convinced that the term 'master regulator' should be applied to trypsin as its mode of action in this scenario remains unknown. I also think that some of the conclusions of the authors are not fully justified by their data and that some further experimental details need to be provided as detailed below.

Response 1: Thank you for your constructive comments on our manuscript. We agree, so far, we still do not know the detailed working mechanisms of PtTryp2 mediated integrating the N–P interactions, and whether it is a master regulator or just one of the major regulators is not so clear. We have decided to remove “master” from the title. We also have done more experiments, with more mutant clones, and new results are consistent with our original results, bolstering our conclusions. In addition, we have modified terminology throughout the text as appropriate.

Comment 2: Title and line 30. Trypsin is a master regulator. The authors provide convincing evidence that knockout or overexpression of Tryp2 alters the N:P ratios of *Phaeodactylum*. Trypsin is a protease that plays an important role in the breakdown of proteins in human nutrition. It is therefore not clear how the activity of trypsin enables it to coordinate different aspects of the N/P metabolism. Whilst the authors freely acknowledge this, I think the lack of a mechanism precludes the description of Tryp2

as a master regulator. This term is normally used to proteins such as transcription factors that directly control the activity of all the downstream proteins in a specific pathway – this has not been demonstrated.

Response 2: As your suggested, we have modified terminology throughout the text as appropriate, as indicated above.

Comment 3: Line 65 Tryp2 expression is downregulated under N-limiting conditions. Whilst Tryp2 is clearly upregulated under P limitation, the down-regulation under N-limitation is not clear to me. To display up- and down-regulation equally, it is better to show gene expression as log fold change (as in used for transcriptomic data in Fig 3a). In Fig 2c, PtTryp2 expression increases substantially from day 3 to day 5 in N-limited cells (and N-replete control cells), but presumably N will have depleted substantially from day 3 to day 5. I agree that Tryp2 expression in N-limited cells is lower than the control at day 9, but it is still higher than N-limited cells at day 3. More details need to be provided in the figure legend of Fig1c to explain this.

Response 3: As suggested, we have changed the y axis scale and plot P values to show PtTryp2 down-regulation under N-depletion clearer. The reason why we did not show the gene expression as log fold change is that it would ignore the cell growth stage-specific expression variation. As shown in Fig. 2, *PtTryp2* was upregulated under P-depletion and downregulated with P supplement, but exhibited an opposite response to N-depletion. The complicated growth stage- and condition- specific expression patterns of PtTryp2 have resulted from the fact that PtTryp2 simultaneously responded to both N and P conditions.

Comment 4: In the Methods, it states Tryp2 expression was normalized to three reference genes, but in Extended Data Fig 2 expression is shown normalized to a single reference gene (PtTBP). The data in Fig2C is presumably shown relative to the

expression of Tryp2 in control (N+P+) cells on day 3, but this all needs to be described clearly. Replication and errors bars also need to be described.

Response 4: Sorry for the confusion. We normalized our qPCR results with three reference genes but only presented the one normalized to PtTBP, because PtTBP is the most stable reference gene based on the web-based comprehensive tool RefFinder, which integrates currently available major computational program analysis (geNorm, Normfinder, BestKeeper, and the comparative Delta-Ct method). In the revised manuscript, we presented the reference gene stability assessment results in Extended Data Table 3. Moreover, the replication and error bars are explained in Figure legends.

Comment 5: As the authors subsequently propose that Tryp2 controls the N/P ratio, it would be interesting to show how its expression co-varies with the N/P ratio.

Response 5: This is a great idea. As you suggested, we have carried out the gene expression pattern analysis of PtTryp2 across different N/P nutrient ratio cultivation conditions, and found that PtTryp2 expression co-varied with the N/P nutrient ratio (Fig. 7c). Moreover, the time-course analysis showed that PtTryp2's expression fluctuated less at an N/P ratio of 16:1 compared to other N/P ratios. The N/P ratio of 16:1 has been proven as optimal for *P. tricornutum* growth (Extended Data Fig. 11).

Comment 6: Line 91 tryp2 mutants. Three mutants were generated and are all described as frameshift mutants (line 287), but one deletion (18 bp) doesn't appear to introduce a frameshift. Only one mutant was used for all subsequent studies, but it is not clear which one was used (line 96). Was the phenotype of the other mutants investigated? The effect on N and P acquisition would be more convincing if demonstrated in multiple mutants, although the authors are able to show that over-expression provides the converse phenotype.

Response 6: The KO mutant used in the original submission was a hybrid mutant, which included different mutated genotypes, and the relevant physiological verification can be considered the average of multiple mutants. In the revised submission, we added two additional clonal mutants, one with 8bp deletion and the other with 41bp deletion (Fig. 4). All the three KO cell lines showed a significant decrease in the expression level, while the OE cell line showed an elevated gene expression level based on qPCR analysis and Western-blot (Figs. 3a and 4c). Furthermore, physiological experiments were conducted over again for all the multiple mutants (see detail results in main text).

Comment 7: The authors do not show growth data for the mutants. It would be interesting to see if they have a defect in growth under N or P limitation.

Response 7: As you suggested, we have placed the growth data of different mutants across different conditions in Fig. 4e and added related descriptions in revised main text. The results demonstrate that both elevation and reduction of *PtTryp2* expression is accompanied by cell growth repression, evidence that *PtTryp2* has a crucial role in modulating cell growth in response to different N and P conditions.

Comment 8: Line 93 Upregulation of N assimilation. Inactivation of Tryp2 led to upregulation of most N assimilation genes (Fig 2a). Whilst many N assimilation genes are upregulated, the pattern of gene expression is very different from N-limited wild type cells, i.e. the genes important in N-limitation are not upregulated in tryp2 KO – this should be made clear.

Response 8: We have made these statements more clear. The statements have been rewritten as " Transcriptomic data show that *PtTryp2* knockout led to the upregulation of most of the nitrogen assimilation and metabolism genes under both N-depleted and replete conditions (Fig. 5a and Extended Data Fig. 10a). Notably, the expression fold change of most N assimilation and metabolism genes under low N, high P (LNHP)

versus high N, high P (HNHP) conditions were moderated in the *PtTryp2* knockout mutant compared to that in its control (KOC), with the exception of GOGAT, which exhibited larger response to the nutrient changes in KOC (Fig. 5a)".

Comment 9: Line 129 *tryp2* knockout downregulates PSI genes but upregulates SPX genes. This effect isn't very clear in Fig 4a (KO vs. KOC in nutrient replete cells).

Response 9: Yes, the effect is more pronounced under P-depleted conditions. This was potentially because of the native expression of *PtTryp2*, PSI genes and SPX genes in KOC was relatively low under nutrient replete condition.

Comment 10: Line 143 *Tryp2* activates PSI genes. Without a clear indication of mechanism, I don't know the authors should state that *Tryp2* functions to activate PSI genes. This infers a direct mode of action of *Tryp2* on these genes that have not been demonstrated.

Response 10: We changed the wording from “activate” to “upregulate”.

Comment 11: Line 165 Abolishment of Pi uptake repression. There is very little repression of Pi uptake shown in the wild type (KOC), with very similar rates of Pi uptake in N-replete versus N-limited cells.

Response 11: We re-measured the relevant physiological parameters of all mutants at 16 hours of culture (the original version was 12 hours). As shown in Fig. 4b, a significant repression of Pi uptake by N-depletion was found in KOC. This indicates that the repression level of Pi or NO_3^- uptake rate increased with the extension of the cultivation time. In the revised version, we describe the experimental method more clearly and in more detail.

Comment 12: Fig 3 and 4. When the cells were harvested for transcriptomic analysis is not made clear. In the Methods, the sentence beginning line 366 does not make sense ('Cells were pre-starvation following the steps:'). Describing the nutrient status of the cells is really important for the interpretation of the transcriptomic data.

Response 12: We are sorry we didn't describe it clearly. In revised manuscript, we have made clearer about the transcriptome sampling and pretreatment in Method sections. We sampled the cells for transcriptomic analysis when the cultures started to show growth depression under N-depleted and P-depleted conditions based on cell growth curve. Considering that the stock culture was kept in f/2 medium ($882\mu\text{M NO}_3^-$ and $36.2\mu\text{M PO}_4^{3-}$), we pre-conditioned a pre-experiment master culture to N-depleted and P-depleted condition (pre-starvation). The starved culture was then used in subsequent experimental culture set up by providing varying nutrient combinations.

Comment 13: Line 44 The term Plantae is usually used to refer to the Archaeplastida. Most phytoplankton do not belong to the group.

Response 13: As suggested, we have corrected it as "the major contributor of marine biodiversity and global CO₂ fixation and O₂ production."

Comment 14: Line 82, Figure 1. The localization of Tryp2 in the chloroplast is not particularly clear in the figure. Is a transmitted light image also available? It would help to show the cytosolic-localized GFP control (Extended data 4).

Response 14: We have reimaged the cell lines that stained with different concentration dyes, different staining times and different negative controls (Fig. 3, Extended Data Figs. 7, 8), and acquired both fluorescence and transmitted light images. Moreover, we have presented the transmitted light image in Figures.

Comment 15: Fig 1c and throughout. The labels N-P+ need to be much clearer. Showing N- or N+ in superscript is very difficult to distinguish in the figures.

Response 15: As suggested, we have replaced N+P+, N-P+ and N+P- with HNHP, LNHP, and HNLP, respectively.

Comment 16: Fig 3a and 4a – N and P assimilation genes in these figures should be labeled with identifiers (as in Extended data 5).

Response 16: We have revised them as suggested.

REVIEWERS' COMMENTS

Reviewer #1 (Remarks to the Author):

I am happy with the revisions performed, and approve this manuscript for publication following the implementation of the subsequent minor amendments:

Line 108: should be "we analysed the physiology of homologous overexpression and CRISPR/Cas9 knockout lines generated in-lab."

Line 110: should be "at a protein level"

Line 253: replace "the phytoplankton experiences" with "Phaeodactylum cells experience"

Lines 287-288: delete "pathway"

Lines 291-294: have the authors crossreferenced their coregulated gene list with previously published datasets generated under non-N and P conditions, e.g. DiatomPortal and PhaeoNet?

Reviewer #2 (Remarks to the Author):

The authors have largely addressed my concerns in their revised manuscript, in many places adding substantial new data. In particular, they have added characterisation of additional clonal tryptophan mutants demonstrating a consistent phenotype. I only have a few additional comments (numbered from my previous review).

Comment 2: Although the authors have removed the term 'master regulator' and replaced it with 'coordinate regulator' I am not wholly convinced that this term is appropriate as it implies that Tryp2 itself is directly controlling the N and P assimilation pathways. Manipulation of Tryp2 expression through knockouts or overexpression clearly has a major impact on N/P metabolism, but the mode of action of trypsin in this pathway remains unknown. One potential explanation could be that deletion/overexpression of the Tryp2 protease interferes with N recycling from protein degradation, leading to an imbalance of cellular N:P. I think this is distinct from a role as a 'master or coordinate regulator', which infers direct control of downstream effectors by Tryp2, although it would still indicate that Tryp2 has an important and novel role in regulating N+P metabolism in diatoms. I suggest that authors expand the Discussion (e.g. paragraph beginning 268) to clarify that the mode of action of Tryp2 remains unknown and that it could contribute to N:P regulation in multiple ways.

Comment 5: The authors state that Tryp2 expression co-varies with N:P (line 243), but I don't see any clear patterns in the data presented in Fig 7c, other than that the expression at N:P 16:1 is relatively stable. Do they mean that deviations from 16:1 result in a low Tryp2 expression initially (4h), and then increased expression after 72 h? If so, this should be explained in the text.

Comment 7: The authors have included growth data (Fig 4e) which shows some interesting effects Tryp2 gene knock out or overexpression. However, in the text (lines 141-150) they need to be careful not to compare growth rates in the non-exponential part of the curve (e.g. 5d-8d) as here low growth rate may be due to the extent to which N or P has been depleted (e.g. control cells may have grown to a higher cell density and stopped growing, this is particularly noticeable in HNLP OEC vs OE). The authors should compare exponential growth rates (d1-d4) only, although it is also valid to comment on differences in the maximum cell density reached in batch culture (as this is effectively a measure of the minimum cell quota of a limiting nutrient). The growth data suggests that tryptophan mutants have a strong growth defect in low P, the authors could present a graph of specific growth rates (d1-d4) to emphasise this effect.

Comment 14: The example shown in Fig 3b to illustrate localisation of Tryp2 in the ER is not

particularly convincing as the ER overlaps almost entirely with the chloroplast in this particular cell. However, there are other examples in Extended Data Fig 7 that show the potential localisation in the ER better. I suggest the authors use a different example for Fig 3.

Responses to the comments from reviewers

Reviewer #1:

I am happy with the revisions performed, and approve this manuscript for publication following the implementation of the subsequent minor amendments:

Comment 1: Line 108: should be “we analysed the physiology of homologous overexpression and CRISPR/Cas9 knockout lines generated in-lab.”

Response 1: We have revised the text as you suggested.

Comment 2: Line 110: should be “at a protein level”

Response 2: We have revised the text as you suggested.

Comment 3: Line 253: replace “the phytoplankton experiences” with “Phaeodactylum cells experience”

Response 3: We have revised the text as you suggested.

Comment 4: Lines 287-288: delete “pathway”

Response 4: We have revised the text as you suggested.

Comment 5: Lines 291-294: have the authors crossreferenced their coregulated gene list with previously published datasets generated under non-N and P conditions, e.g. DiatomPortal and PhaeoNet?

Response 5: Thank you for your constructive suggestions. We now have conducted a comparative analysis for the potential co-regulated gene list identified in this study with the published co-regulatory analysis datasets DiatomPortal and PhaeoNet. Interestingly, based on DiatomPortal dataset, the *PtTryp2* was found in the Phatr_hclust_0381 hierarchical cluster that consisted of 10 genes, which has been identified as the GO term of ubiquitin-dependent protein catabolism. In terrestrial plants, the ubiquitination and degradation of SPX4 was found to mediate the nitrate–phosphate interaction signaling

pathway, that is the ubiquitination and degradation of SPX4 enables the subsequent release of PHR2 and NLP3 into the nucleus to activate the expression of both phosphate- and nitrate-responsive genes. In addition, we found 120 genes that were common between our gene list and that from PhaeoNet, some of which are transcription factors. Taken together, the simultaneous impact in opposite directions of *PtTrp2* on N and P suggests that this protein either directly regulate N and P uptake machinery or is close to that direct regulator, e.g. potentially functioning through ubiquitination and degradation of direct regulators as in terrestrial plants. This provides new clues for future studies to unravel the exact regulatory mechanism. We have added the information to the manuscript (p13, lines 365-374)

Reviewer #2:

The authors have largely addressed my concerns in their revised manuscript, in many places adding substantial new data. In particular, they have added characterization of additional clonal *tryp2* mutants demonstrating a consistent phenotype. I only have a few additional comments (numbered from my previous review).

Comment 1: Although the authors have removed the term ‘master regulator’ and replaced it with ‘coordinate regulator’ I am not wholly convinced that this term is appropriate as it implies that *Tryp2* itself is directly controlling the N and P assimilation pathways. Manipulation of *Tryp2* expression through knockouts or overexpression clearly has a major impact on N/P metabolism, but the mode of action of trypsin in this pathway remains unknown. One potential explanation could be that deletion/overexpression of the *Tryp2* protease interferes with N recycling from protein degradation, leading to an imbalance of cellular N:P. I think this is distinct from a role as a ‘master or coordinate regulator’, which infers direct control of downstream effectors by *Tryp2*, although it would still indicate that *Tryp2* has an important and novel role in regulating N+P metabolism in diatoms. I suggest that authors expand the Discussion (e.g. paragraph beginning 268) to clarify that the mode of action of *Tryp2* remains unknown and that it could contribute to N:P regulation in multiple ways.

Response 1: We appreciate your insight. We have thought about the possibility you provided, which is intuitive because trypsin is known as a protease. However, our data showed that the deletion and overexpression of *PtTryp2* simultaneously impacted nitrogen and phosphorus uptake, nitrogen and phosphorus contents of the cell, and N:P (Figs. 5-7). Furthermore, the response of *PtTryp2* expression to P addition was much more dramatic than to N addition (Fig. 2). These indicate that the role of this gene in N/P balance is not likely due to its impact on N recycling from proteins. More importantly, the simultaneous impact of *PtTryp2* in opposite directions on N and P suggests that this protein either directly regulates the N and P uptake machinery or is close to that direct regulator. Moreover, in the revised manuscript, we have conducted a comparative analysis for the potential co-regulated gene list identified in this study with the published co-regulatory analysis datasets DiatomPortal and PhaeoNet. Interestingly, based on DiatomPortal dataset, the *PtTryp2* was found in the Phatr_hclust_0381 hierarchical cluster that consisted of 10 genes, which has been identified as the GO term of ubiquitin-dependent protein catabolism. In terrestrial plants, the ubiquitination and degradation of SPX4 was found to mediate the nitrate–phosphate interaction signaling pathway, that is the ubiquitination and degradation of SPX4 enables the subsequent release of PHR2 and NLP3 into the nucleus to activate the expression of both phosphate- and nitrate-responsive genes. In addition, we found 120 genes that were common in our gene list and PhaeoNet, some of which are transcription factors. Taken together, the current data and literature suggest in concert that *PtTryp2* either directly regulates N and P uptake machinery or is close to that direct regulator, e.g., potentially functioning through ubiquitination and degradation of direct regulators as in terrestrial plants. Furthermore, we believe, that one or more intermediate relays between *PtTryp2* and the direct regulator would make it extremely challenging, if not impossible, to exert such precise bidirectional regulation. We have added this discussion to the manuscript (page 13, lines 365-383).

Comment 2: The authors state that Tryp2 expression co-varies with N:P (line 243), but I don't see any clear patterns in the data presented in Fig 7c, other than that the

expression at N:P 16:1 is relatively stable. Do they mean that deviations from 16:1 result in a low *Tryp2* expression initially (4h), and then increased expression after 72 h? If so, this should be explained in the text.

Response 2: In Fig. 7C, we meant to show that *PtTryp2* expression was relatively stable at N:P=16:1 because 16:1 is the optimal N:P ratio for *Phaeodactylum tricornutum* growth, needing no significant changes in *PtTryp2* expression to maintain N/P balance, but other N:P ratio nutrient conditions deviating from 16:1 caused changes in *PtTryp2* expression to maintain N/P balance. Moreover, the change was different between different levels of nutrient supply N:P ratios, and between 4h and 72h after nutrient addition. We are sorry that we did not make it clear in the text but 4h represents nutrient-repletion and 72 h nutrient-depletion. We now have added the information in the figure legend. At 72h, the expression level of *PtTryp2* increased with the degree of P stress (the higher the N:P ratio, the more P stressed the cultures were), except that at 1:1 of N:P, an extreme N-limited condition that seemed to cause *PtTryp2* expression response not accordingly to the general trend. Overall, all these data indicate that *PtTryp2* responds to N:P ratio.

Comment 3: The authors have included growth data (Fig 4e) which shows some interesting effects *Tryp2* gene knock out or overexpression. However, in the text (lines 141-150) they need to be careful not to compare growth rates in the non-exponential part of the curve (e.g. 5d-8d) as here low growth rate may be due to the extent to which N or P has been depleted (e.g. control cells may have grown to a higher cell density and stopped growing, this is particularly noticeable in HNLP OEC vs OE). The authors should compare exponential growth rates (d1-d4) only, although it is also valid to comment on differences in the maximum cell density reached in batch culture (as this is effectively a measure of the minimum cell quota of a limiting nutrient). The growth data suggests that *tryp2* mutants have a strong growth defect in low P, the authors could present a graph of specific growth rates (d1-d4) to emphasise this effect.

Response 3: This is a great idea. As your suggested, we have presented a graph of specific growth rates (d1-d4) in the Extended Fig. 9, and made more accurate

description of growth status based on exponential growth rates and the maximum cell density in the revised manuscript.

Comment 4: The example shown in Fig 3b to illustrate localisation of Tryp2 in the ER is not particularly convincing as the ER overlaps almost entirely with the chloroplast in this particular cell. However, there are other examples in Extended Data Fig 7 that show the potential localisation in the ER better. I suggest the authors use a different example for Fig 3.

Response 4: As your suggested, we have used another example from Extended Data Fig 7 for Fig 3b ER panel in revised version.